# Are you **SURE**? Enhancing Multimodal Pretrained Frameworks upon Missing Modalities via Uncertainty Estimation

## Abstract

Multimodal learning has achieved remarkable success by integrating diverse data sources, yet it often assumes the availability of all modalities - an assumption rarely met in real-world settings. While pretrained multimodal models are powerful, they struggle with small-scale and incomplete datasets (i.e., missing modalities), which limits their practical utility. Previous work on reconstructing missing modalities has largely ignored the potential unreliability of these reconstructions, risking the quality of final predictions. We propose **SURE** (Scalable Uncertainty and Reconstruction Estimation), a framework that enhances pretrained multimodal models by introducing latent space reconstruction and robust uncertainty estimation for both reconstructed modalities and downstream tasks. Our framework not only enhances performance but also offers a reliable uncertainty metric, improving interpretability. Key innovations include a novel Pearson Correlation-based loss and the first application of statistical error propagation in deep networks, enabling precise uncertainty quantification from missing data and model predictions. Extensive experiments on tasks like sentiment analysis, genre classification, and action recognition demonstrate that SURE consistently achieves state-of-the-art performance, offering robust predictions even with incomplete data.

## 1 Introduction

**Motivation:** Multimodal learning has proven to be an effective approach for handling raw data from various sources and formats, often outperforming traditional unimodal learning techniques Huang et al. (2021). However, despite achieving state-of-the-art performance across various tasks Zong & Sun (2023); Wan et al. (2023); Wu et al. (2024), most leading multimodal frameworks rely on idealized conditions during training and evaluation, assuming access to a great volume of data and all modalities are available. Nonetheless, such ideal conditions often break down in real-world applications (e.g. autonomous vehicles or medical centers).

To deal with limited dataset scale, leveraging models pre-trained on larger datasets for similar tasks has emerged as a promising and efficient solution. While this approach is extensively utilized in unimodal settings He et al. (2016); Devlin (2018), it remains underexplored in multimodal contexts. To showcase the benefit of pretrained weights, we compared two versions of MMML model Wu et al. (2024), a state-of-the-art fusion architecture for Semantic Analysis, on the CMU-MOSI dataset Zadeh et al. (2016). One model was initialized with pretrained weights from the larger CMU-MOSEI dataset Zadeh et al. (2018), while the other was trained from scratch. Performance was evaluated across varying training set sizes, scaled proportionally to the original CMU-MOSI dataset (Figure 1). The observed performance gap underscores the significant potential of leveraging pretrained models to enhance both training effectiveness and efficiency on small-scale datasets.

A significant barrier to the widespread adoption of pretrained multimodal frameworks is their *inability to handle missing modalities during training or evaluation*. These models typically assume the availability of all modalities and struggle to adapt effectively when some are absent. Furthermore, even when they manage to function with incomplete data, they often lack mechanisms to *assess the reliability of their predictions* in such cases. This limitation inevitably leads to degraded performance compared to scenarios with full modality availability, as these models are trained under

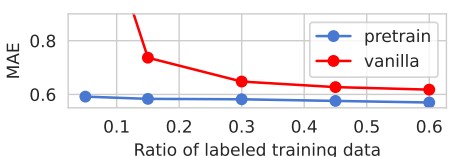

Figure 1: Comparison of pretrained and vanilla MMML framework on CMU-MOSI dataset.

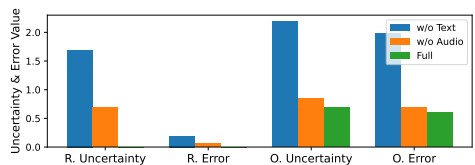

Figure 2: SURE reconstructs missing modalities for final predictions, reporting average errors and uncertainties for both reconstruction and output.

ideal conditions. This challenge is especially critical in safety-sensitive domains like healthcare and autonomous driving, where robust and trustworthy decision-making is paramount. Incorporating uncertainty estimation becomes essential in these scenarios to gauge prediction reliability and mitigate risks. The last two bar groups in Figure 2 illustrates this issue using our model's output on the CMU-MOSI evaluation dataset under different input modality combinations. The figure records the average values of Error (MSE) and Estimated Uncertainty. As shown, missing modalities degrade the model's performance, particularly when text is absent. However, the model also generates reasonable uncertainty estimates for its predictions. Notably, higher average uncertainties align with higher errors, suggesting that uncertainty estimates can effectively serve as indicators of prediction reliability in case of missing modalities.

**Current Literature:** The first challenge we highlight, missing modalities, is a prevalent issue in the training and deployment of multimodal models. To address this problem, research has focused on two main approaches: (1) Contrastive loss-based strategies that align latent spaces for cross-modal knowledge transfer Ma et al. (2022); Lee & Van der Schaar (2021); Wang et al. (2020), and (2) generative strategies, such as VAE variations Wu & Goodman (2018) or latent space reconstruction Woo et al. (2023), to recreate missing modalities. The latter approach can be useful to incorporate with pretrained multimodal frameworks, as it introduces modules or techniques to reconstruct missing modalities without altering the remaining parts of the models.

To assess model reliability, a growing branch of research focuses on estimating uncertainty in predictions. Common directions include Bayesian deep learning Wang et al. (2019b); Kendall & Gal (2017a), which models uncertainty directly by estimating the output distribution, and post-hoc techniques Maddox et al. (2019); Lakshminarayanan et al. (2017) that introduce perturbations to the inputs to generate multiple outputs, enabling uncertainty estimation. Adapting or extending these methods offers valuable support for decision-making during evaluation.

**Our approach:** Inspired by existing literature, this work tackles the real-world challenges of small-scale datasets with missing modalities through two key contributions: (1) the effective utilization of pretrained multimodal frameworks with a simple latent space reconstruction strategy, and (2) the estimation of reconstruction and output uncertainties, emphasizing their interdependence and relationship with downstream task performance. Specifically, we show that higher reconstruction uncertainty correlates with higher output uncertainty, which in turn leads to higher output error (Figure 2). To achieve this, we focus on three types of uncertainties:

(1) Reconstruction uncertainty for the missing modalities.
(2i) Output uncertainty stemming from the reconstructed inputs.
(2ii) Output uncertainty arising from the inherent nature of the model.

For uncertainties (1) and (2ii), we propose a novel loss function based on Pearson Correlation, combined with a tailored training strategy that balances downstream task optimization with uncertainty estimation (see Section 2.3). To estimate uncertainty (2i), we introduce the first application of Error Propagation Arras (1998); Tellinghuisen (2001) in deep neural network training (see Section 2.4).

The proposed approach, named **SURE** (Scalable Uncertainty and Reconstruction Estimation), can be integrated seamlessly with any pretrained multimodal deep network. This flexibility allows it to handle datasets with missing modalities while estimating both input and output uncertainties. Importantly, our uncertainty estimation mechanisms empower the model to recognize when it faces

 In summary, the key contributions are summarized as follows:

- We introduce SURE, a pipeline that leverages pretrained models and estimates reconstruction and output uncertainties for scenarios with missing modalities.

- We explicitly model the interconnections between reconstruction uncertainty, output reliability, and downstream task performance.

- We achieve new state-of-the-art results across all three downstream tasks while effectively estimating the corresponding uncertainties.

## 2 PROPOSED METHOD

### 2.1 PROBLEM FORMULATION

Let $D = \bigcup_i \{(\mathbf{x}_i, \mathbf{y}_i)\}$ be the training dataset with pairs from domain $\mathcal{X}$ and $\mathcal{Y}$ where $\mathcal{X}, \mathcal{Y}$ lies in $\mathbb{R}^{n_1} \times ... \times \mathbb{R}^{n_M}$ and $\mathbb{R}^k$ respectively. Let denote $\mathbf{x}_i = (\mathbf{x}_i^1, ..., \mathbf{x}_i^M)$ is the $i^{th}$ input sample where $\mathbf{x}_i^j$ is its $j^{th}$ modality. We consider scenarios in which during training or evaluation, certain modalities are missing in certain samples.

### 2.2 OVERVIEW OF SURE FRAMEWORK

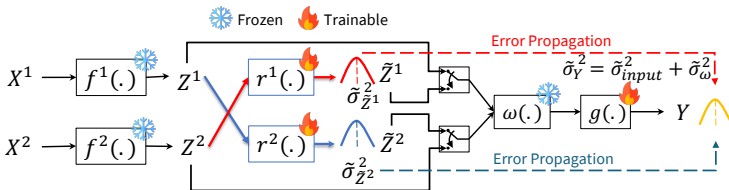

Figure 3: Overview of proposed module. It incorporates a set of reconstruction modules $r^i(.)$' in the middle layers of arbitrary pretrained multimodal fusion frameworks, after their latent space projection layers $f^i(.)$ and before their fusion layers $\omega(.)$. These modules consume other modality latent representation, yield reconstructed representation for the corresponding modality, together with reconstruction uncertainty. The reconstructed output replaces the role of the original missing modality, while the uncertainty is propagated through the rest of the network to capture output uncertainty caused by the reconstruction. The final classifier additionally involves the calculation of output uncertainty caused by model's inherent nature.

The overall proposed pipeline is depicted in Figure 3. For clarity and simplicity, we illustrate the SURE pipeline using two modalities, with a straightforward extension to $M$ modalities discussed in Appendix A.1.3. In our approach, the only components that require training are the reconstruction modules $r^i(.)$, each tailored for a specific modality, and the final classifier head $g(.)$. This configuration preserves most of the pretrained framework, with the exception of the final classifier heads, aligning with standard procedures used in unimodal pretraining and finetuning. In this study, two primary objectives are established to fully leverage the capabilities of pretrained models: (1) **reconstructing missing modalities**, and (2) **assessing the reliability of the final outputs**.

To address the first objective, we introduce an efficient reconstruction procedure within the shared latent space. Let $Z^i$ denote the latent representation for each modality produced by the pretrained framework's unimodal projectors ($f^i(.)$ in Fig.3). SURE incorporates reconstruction modules, $r^i(.)$, each specifically designed for a particular modality. The reconstructor for the $i^{th}$ modality, $r^i(.)$, utilizes the latent representation of another available modality (e.g., $Z^j$) to generate an approximation $\tilde{Z}^i$ in the event that the $i^{th}$ modality is missing, e.g., $r^i(Z^j) = \tilde{Z}^i$ $(i = 1, \ldots, M; j \neq i)$. Additional details and analysis regarding these modules are discussed in Appendix A.1.2.

The second objective naturally arises from the presence of reconstructed inputs, making it crucial to assess the reliability of these reconstructions, as well as the final results produced by the model using them. To this end, we have designed both the reconstruction modules and the final classifier to

produce probabilistic outputs that accurately reflect the associated uncertainties. Specifically, when reconstructing the representation of the $i^{th}$ modality $\tilde{Z}^i$ using another modality's representation $Z^j$, the module $r^i(.)$ is tailored to approximate the underlying distribution $\mathcal{P}_{\tilde{Z}^i|Z^j}$. Similarly, the classifier head aims to produce the probabilistic outcome $\mathcal{P}_{Y|X}$. Although the exact nature of these distributions is unknown, we introduce a novel yet straightforward loss function that effectively quantifies the uncertainty associated with the reconstructed input and the output uncertainty arising from the stochastic nature of the pretrained framework (Section 2.3). Additionally, we quantify the uncertainty that propagates through the framework due to the uncertainty associated with the reconstructed input, as detailed in Section 2.4.

## 2.3 Distribution-free Uncertainty Estimation

**Uncertainty Estimation Preliminaries.** Bayesian deep learning models are capable of capturing uncertainty in their outputs. By maximizing a likelihood function, these models can optimize the network parameters to estimate the output distribution. This allows them to effectively capture aleatoric uncertainty, which arises from the inherent noise in the input data. Specifically, for a given input $\mathbf{x}_i$ from input domain $\mathcal{X}$, the corresponding model output $\{\tilde{\mathbf{y}}_i, (\tilde{\zeta}_i^1, \ldots, \tilde{\zeta}_i^k)\} := \phi(\mathbf{x}_i, \theta)$ are parameters to specify the distribution $\mathcal{P}_{\mathbf{Y}|\mathbf{X}}$. The likelihood $\mathcal{L}(\theta, \mathcal{D}) := \prod_{i=1}^N p_{\mathbf{Y}|\mathbf{X}}(\mathbf{y}_i; \{\tilde{\mathbf{y}}_i, \tilde{\zeta}_i^1, \ldots, \tilde{\zeta}_i^k\})$ is then maximized to estimate the optimal parameters of the model. Additionally, the distribution $\mathcal{P}_{\mathbf{Y}|\mathbf{X}}$ is often chosen so that uncertainty can be estimated using a closed-form solution based on the model's estimated parameters. A common choice for this distribution is the heteroscedastic Gaussian Upadhyay et al. (2022); Wang et al. (2019b); Kendall & Gal (2017a), where $\phi(., \theta)$ predicts both the mean and variance, e.g. $\{\tilde{\mathbf{y}}_i, \tilde{\sigma}_i\} := \phi(\mathbf{x}_i, \theta)$. In this context, the predicted variance can be interpreted as the uncertainty of the prediction. The optimization problem then becomes:

$$\theta^* = \arg\max_\theta \prod_{i=1}^N \frac{1}{\sqrt{2\pi\tilde{\sigma}_i^2}} e^{-\frac{\|\tilde{y}_i - y_i\|^2}{2\tilde{\sigma}_i^2}} = \arg\min_\theta \sum_{i=1}^N \frac{\|\tilde{\mathbf{y}}_i - \mathbf{y}_i\|^2}{2\tilde{\sigma}_i^2} + \frac{\log(\tilde{\sigma}_i^2)}{2} \tag{1}$$

$$\text{Uncertainty}(\tilde{\mathbf{y}}_i) = \tilde{\sigma}_i^2. \tag{2}$$

The closed-form solutions $\tilde{\mathbf{y}}^i$ and $\tilde{\sigma}_i^2$ for the problem in Equation 1 can be derived (the detailed process is shown in Appendix A.1.1), with the final results as follows:

$$\tilde{\mathbf{y}}_i^* = \mathbf{y}_i; \quad \tilde{\sigma}_i^{2*} = \tilde{\epsilon}_i^2 \text{ where } \tilde{\epsilon}_i^2 := \|\tilde{\mathbf{y}}_i - \mathbf{y}_i\|^2. \tag{3}$$

The main drawback of this common approach is its reliance on assumptions about the true underlying distribution $\mathcal{P}_{\mathbf{Y}|\mathbf{X}}$, which may not hold true across all datasets and scenarios. Additionally, estimating uncertainty based specifically on a Gaussian assumption presents a further limitation due to the strict closed-form solution for $\tilde{\sigma}_i^{2*}$ (Equation 3). Ideally, the objective of Equation 1 would aim to learn a model capable of precisely approximating its own error $\tilde{\epsilon}^2$. However, this becomes increasingly difficult as $\tilde{\epsilon}^2 \to 0$, leading to unstable loss values and poorly defined gradients with respect to $\tilde{\sigma}_i^{2*}$ (detailed explanation in Appendix A.1.1). Consequently, achieving this strict objective through gradient-based optimization is highly challenging, if not impossible.

**Our method.** To overcome these limitations, we introduce a straightforward, distribution-free loss function centered on a more adaptable constraint for uncertainty: *ensuring a strong correlation with error.* This loss function are employed to learn both reconstruction uncertainty as well as output uncertainty owing to model inherent nature. Specifically, we design our loss to learn the uncertainty $\tilde{\sigma}^2$ by leveraging the Pearson Correlation Coefficient Cohen et al. (2009) between $\tilde{\sigma}^2$ and prediction error $\tilde{\epsilon}^2$:

$$\mathcal{L}_{PCC}(\tilde{\sigma}^2, \tilde{\epsilon}^2) = 1 - r(\tilde{\sigma}^2, \tilde{\epsilon}^2);$$

$$\text{where } r(\tilde{\sigma}^2, \tilde{\epsilon}^2) = \frac{\sum_{i=1}^N (\tilde{\sigma}_i^2 - \mu_{\sigma^2})(\tilde{\epsilon}_i^2 - \mu_{\epsilon^2})}{\sqrt{\sum_{i=1}^N (\tilde{\sigma}_i^2 - \mu_{\sigma^2})^2} \sqrt{\sum_{i=1}^N (\tilde{\epsilon}_i^2 - \mu_{\epsilon^2})^2}},$$

$$\mu_{\sigma^2} = \frac{1}{N} \sum_{i=1}^N \tilde{\sigma}_i^2, \mu_{\epsilon^2} = \frac{1}{N} \sum_{i=1}^N \tilde{\epsilon}_i^2, \tilde{\epsilon}_i^2 = \|\tilde{\mathbf{y}}_i - \mathbf{y}_i\|^2, \tag{4}$$

$N$ is the number of samples in one batch. The Pearson correlation coefficient is essentially a normalized measure of covariance, ensuring that the result always falls within the range of -1 to 1.

Consequently, $\mathcal{L}_{PCC}$ is constrained to a value between 0 and 2. A value of $\mathcal{L}_{PCC} = 0$ indicates a perfect linear relationship where $\tilde{\sigma}^2$ increases in tandem with $\tilde{\epsilon}^2$, meaning the uncertainty accurately reflects the prediction error. Conversely, $\mathcal{L}_{PCC} = 2$ implies an inverse relationship. This metric focuses on the linear correlation between squared error and uncertainty while relaxing constraints on the magnitude of the features.

In essence, $\mathcal{L}_{PCC}$ is equivalent to the Mean Squared Error (MSE) between squared error and uncertainty after standardization. Let $\bar{\sigma}_i^2 := \frac{\tilde{\sigma}_i^2 - \mu_\sigma}{\sqrt{\frac{1}{N-1}\sum_{j=1}^N (\tilde{\sigma}_j^2 - \mu_{\sigma^2})^2}}$ and $\bar{\epsilon}_i^2 := \frac{\tilde{\epsilon}_i^2 - \mu_{\epsilon^2}}{\sqrt{\frac{1}{N-1}\sum_{j=1}^N (\tilde{\epsilon}_j^2 - \mu_{\epsilon^2})^2}}$ be the standardized version of $\tilde{\sigma}^2$ and $\tilde{\epsilon}^2$ within a mini-batch, ensuring they have zero mean and unit variance. Given that $\frac{1}{N-1}\sum_i (\bar{\sigma}_i^2)^2 = 1$ and $\frac{1}{N-1}\sum_i (\bar{\epsilon}i^2)^2 = 1$, we have the derivation as follow:

$$\frac{1}{2N}\sum_{i=1}^N (\bar{\sigma}_i^2 - \bar{\epsilon}_i^2)^2 = \frac{1}{2N}\left((2N-2) - 2\sum_{i=1}^N \bar{\sigma}_i^2 \bar{\epsilon}_i^2\right) = \frac{2N-2}{2N}(1 - r(\tilde{\sigma}^2, \tilde{\epsilon}^2)) \approx \mathcal{L}_{PCC}. \quad (5)$$

This equivalence suggests that the loss function relaxes constraints on the magnitude of both $\tilde{\epsilon}^2$ and $\tilde{\sigma}^2$, while still enforcing a linear dependency between these two variables. After training with this loss function, the output uncertainty can be scaled using the mean and standard deviation of the training errors, allowing it to be approximately aligned with the actual testing error. Additionally, we later show in Appendix A.1.1 that our loss function promotes a more stable training process near the optimal solution, which is not the case for ordinary Gaussian NLL loss.

In addition to uncertainty estimation, other loss functions are employed to learn the reconstruction or output corresponding to the downstream task. The estimated uncertainty is aligned with the corresponding reconstruction or prediction error. For reconstruction or regression tasks, Mean Squared Error (MSE) is used to quantify the error, while for classification tasks, the error is represented by the cross-entropy loss value. For instance, final loss guiding SURE's reconstruction modules is:

$$\mathcal{L}_{rec}^i(\tilde{\mathbf{z}}^i, \tilde{\sigma}_{z^i}^2) = \frac{1}{N}\sum \|\tilde{\mathbf{z}}^i - \mathbf{z}^i\|^2 + \mathcal{L}_{PCC}(\tilde{\sigma}_{z^i}^2, \|\tilde{\mathbf{z}}^i - \mathbf{z}^i\|^2) \quad (6)$$

Here, the first MSE term guides the learning of the reconstructed $\tilde{\mathbf{z}}^i$, while $\mathcal{L}_{PCC}$ directs the learning of $\sigma^2$ to accurately reflect the reconstruction error. A similar loss function is applied to estimate the uncertainty of the output, accounting for the stochastic nature of pretrained models. In this context, the error $\tilde{\epsilon}^2$ is defined as MSE for regression tasks or Cross Entropy for classification tasks. In parallel, we enhance the estimated output uncertainty by quantifying the uncertainty propagated from the reconstructed input, utilizing Error Propagation through the frozen pretrained network, as detailed in Section 2.4.

## 2.4 ERROR PROPAGATION THROUGH DEEP NETWORKS

**Error Propagation Preliminaries.** While employing PCC loss in the classifier head effectively models output uncertainty due to the stochastic nature of pretrained frameworks, a strategy is still needed to quantify uncertainty related to the reconstructed input. Error propagation Arras (1998); Tellinghuisen (2001) is a fundamental concept in scientific measurement and data analysis, exactly used to quantify how uncertainties in input variables affect the uncertainty in a derived quantity. Given a function $f(A_1, A_2, \ldots, A_n)$ of $n$ variables, each attached with uncertainty $\sigma_{A_i}^2$ ($i = 1, \ldots, n$), the total uncertainty in the function's output, denoted as $\sigma_f^2$, is determined by how these input uncertainties propagate through the function. This quantity is then calculated using the formula Arras (1998); Tellinghuisen (2001):

$$\sigma_f^2 \approx \sum_{i=1}^n \left(\frac{\partial f}{\partial A_i}\right)^2 \sigma_{A_i}^2, \quad (7)$$

where $\frac{\partial f}{\partial A_i}$ represents the partial derivative of the function with respect to $A_i$. This formula assumes that the uncertainties $\sigma_{A_i}^2$ ($i = 1, \ldots, n$) are independent and uncorrelated.

**Our Method.** This concept aligns perfectly with our goal of quantifying output uncertainty propagated from reconstructed inputs. Therefore, we adapt it to SURE's pretrained deep model pipeline. Given SURE's pretrained model, denoted as $\omega(\{Z^i\}_{i \in \mathcal{I}}, \{\tilde{Z}^j\}_{j \in \mathcal{J}})$ where $\mathcal{I}$ and $\mathcal{J}$ represent the

sets of indices for which $Z^i$ is available or unavailable, respectively, the corresponding output uncertainty propagated from the reconstructed input is:

$$\tilde{\sigma}^2_{input} \approx \sum_{i \in \mathcal{J}} \left( \frac{\partial \omega}{\partial \tilde{Z}_i} \right)^2 \sigma^2_{\tilde{Z}_i}. \tag{8}$$

Combining $\tilde{\sigma}^2_{input}$ with the uncertainty stem from $\omega(.)$ stochastic nature, denoted by $\tilde{\sigma}_\omega$ (learnt with $\mathcal{L}_{PCC}$ - Section 2.3), we achieve the final output uncertainty:

$$\tilde{\sigma}^2_Y = \tilde{\sigma}^2_{input} + \tilde{\sigma}^2_\omega. \tag{9}$$

This combination follows the Pythagorean theorem for variances, which assumes the estimated uncertainty are caused by individual sources Dieck (2007), which is suitable for our use case.

**Training process.** In the initial phase, the reconstruction module is trained using $\mathcal{L}_{rec}$. One of the available modalities is used as the ground-truth output, while the remaining modalities serve as input for prediction. In the second phase, all reconstruction modules are frozen, and the classifier head is trained using $\mathcal{L}_{downstream}$. A detailed summary of the training process is provided in Algorithm 1 in Appendix A.1.3.

## 3 EXPERIMENTS

### 3.1 DATASETS AND METRICS

We integrate SURE into three pretrained frameworks, adapting them to smaller-scale datasets with missing modalities during training and testing. Additional integration details are covered in Appendix A.1.4.

**Sentiment Analysis.** This task involves predicting the polarity of input data (e.g., video, transcript). We use the state-of-the-art multimodal architecture, MMML Wu et al. (2024), pretrained on the CMU-MOSEI dataset Zadeh et al. (2018), which includes video and sound. To assess the SURE + MMML pipeline (referred to as SURE hereafter), we fine-tune it on an incomplete version of the CMU-MOSI dataset Zadeh et al. (2016), where certain modalities are missing during both training and evaluation. Evaluation metrics include mean absolute error (MAE), correlation (Corr), binary Accuracy and F1 score, following Wu et al. (2024); Poklukar et al. (2022); Tsai et al. (2018).

**Book genre classification.** This task involves classifying book genres based on their titles, summaries (text), and covers (images). We integrate SURE with MMBT Kiela et al. (2019), a pretrained framework originally designed for movie genre classification on the MM-IMDB dataset Arevalo et al. (2020). The book dataset used for genre classification is sourced from Haque et al. (2022). Accuracy and F1 scores are used for performance assessment.

**Human Action Recognition.** This task involves identifying human actions based on recorded videos and sensor data. We use HAMLET framework Islam & Iqbal (2020), pretrained on the large-scale MMAct dataset Kong et al. (2019). For prediction, we leverage three modalities: RGB videos, smartwatch acceleration, and phone gyroscope data. Fine-tuning HAMLET + SURE pipeline is done on the smaller, incomplete UTD-MHAD dataset Chen et al. (2015). Performance is evaluated using accuracy and F1 scores.

For all tasks, we evaluate the quality of the estimated uncertainty using two metrics. Uncertainty Calibration Error (UCE) Guo et al. (2017) quantifies the discrepancy between predictive error and uncertainty, while the Pearson Correlation Coefficient (PCC) Upadhyay et al. (2022) measures the correlation between them, with higher values indicating better alignment.

### 3.2 BASELINES AND EXPERIMENT DETAILS

**Baselines.** In our comparative evaluation, we incorporate several state-of-the-art approaches, each representing prominent strategies. The baselines are grouped into two categories, reflecting the key challenges addressed by SURE: (1) Reconstruction methods for missing modalities, and (2) Uncertainty estimation methods. The reconstruction techniques include ActionMAE Woo et al. (2023), DiCMoR Wang et al. (2023), and IMDer Wang et al. (2024). For uncertainty estimation,

we evaluate against the Gaussian Maximum Likelihood method Kendall & Gal (2017b); Wang et al. (2019a), Monte Carlo Dropout Maddox et al. (2019); Laves et al. (2019); Srivastava et al. (2014), and Ensemble Learning Lakshminarayanan et al. (2017). The original codebases of all baseline implementations are used for best reproducibility.

**Experiment details.** For all reconstruction methods, we use the same pretrained frameworks as SURE for fair comparison. Uncertainty estimation methods are integrated into the SURE* + pretrained models pipeline, where SURE* is a deterministic version using MSE loss instead of our proposed $\mathcal{L}_{rec}(.,.)$. Training datasets are modified with $50\%$ of each modality's samples masked as missing, using distinct masks across modalities. Detailed settings are in Appendix A.1.4.

## 3.3 MAIN RESULTS

The reported results include the performance of each pipeline given either unimodal inputs (only a single modality available) or full inputs (all modalities available), averaged over three runs with different random seeds. The best and second-best metrics are highlighted in **red** and blue, respectively.

**Sentiment Analysis.** The results for the CMU-MOSI dataset are summarized in Table 1. SURE and its variations consistently outperform recent reconstruction techniques, highlighting their effectiveness in handling missing modalities. SURE's ability to reconstruct missing data on the fly during training allows every sample to be fully utilized, leading to improved final outputs. Among the modalities, audio appears to be less effective for the downstream task. All methods perform better when text is available compared to when only audio is used, and uncertainty estimation also declines when relying solely on audio.

Table 1: Results of different approaches on CMU-MOSI Dataset.

| Model | MAE | | | Corr | | | F1 | | | Acc | | | Reconstruct Uncertainty Corr | | Output Uncertainty Corr | | | Output UCE | | |
|---|---|---|---|---|---|---|---|---|---|---|---|---|---|---|---|---|---|---|---|---|
| | T(ext) | A(udio) | F(ull) | T | A | F | T | A | F | T | A | F | T | A | T | A | F | T | A | F |
| *Modality Reconstruction Techniques:* | | | | | | | | | | | | | | | | | | | | |
| ActionMAE | 1.106 | 2.146 | 1.005 | 0.506 | 0.155 | 0.517 | 0.717 | 0.57 | 0.719 | 0.724 | 0.423 | 0.725 | - | - | - | - | - | - | - | - |
| DiCMoR | 0.811 | 1.227 | 1.106 | 0.783 | 0.427 | 0.537 | 0.854 | 0.57 | 0.65 | 0.856 | 0.585 | 0.654 | - | - | - | - | - | - | - | - |
| IMDer | 0.707 | 1.237 | 1.106 | 0.797 | 0.438 | 0.544 | 0.846 | 0.524 | 0.62 | 0.846 | 0.564 | 0.634 | - | - | - | - | - | - | - | - |
| *Uncertainty Estimation Techniques:* | | | | | | | | | | | | | | | | | | | | |
| SURE + Gaussian MLE | 0.589 | 1.133 | 0.581 | 0.866 | 0.53 | 0.871 | 0.88 | 0.676 | 0.885 | 0.879 | 0.678 | 0.882 | 0.103 | 0.013 | 0.067 | 0.032 | 0.059 | 0.425 | 0.476 | 0.385 |
| SURE + MC DropOut | 0.63 | 1.153 | 0.622 | 0.858 | 0.556 | 0.865 | 0.877 | 0.686 | 0.899 | 0.876 | 0.684 | 0.9 | 0.047 | 0.008 | 0.013 | 0.009 | 0.13 | 0.496 | 0.51 | 0.396 |
| SURE + DeepEnsemble | 0.592 | 1.071 | 0.582 | 0.868 | 0.58 | 0.871 | 0.886 | 0.714 | 0.889 | 0.885 | 0.716 | 0.888 | 0.062 | 0.031 | 0.024 | 0.074 | 0.082 | 0.497 | 0.492 | 0.389 |
| **SURE** | 0.602 | 1.148 | 0.583 | 0.865 | 0.557 | 0.869 | 0.896 | 0.685 | 0.891 | 0.894 | 0.684 | 0.89 | 0.739 | 0.732 | 0.381 | 0.18 | 0.485 | 0.315 | 0.429 | 0.285 |

**Book Genre Classification.** Similar to the sentiment analysis task, SURE outperforms recent reconstruction techniques in this classification task (Table 2), showing a stronger correlation between uncertainty and error for both reconstruction and downstream tasks. In the Book Dataset, the text modality proves to be highly effective for the downstream task, but it contributes less to uncertainty estimation for both reconstruction and downstream tasks.

Table 2: Results of different approaches on Book Dataset.

| Model | F1 | | | Acc | | | Reconstruct Uncertainty Corr | | Output Uncertainty Corr | | | Output UCE | | |
|---|---|---|---|---|---|---|---|---|---|---|---|---|---|---|
| | T(ext) | I(mage) | F(ull) | T | I | F | T | I | T | I | F | T | I | F |
| Uncertainty Estimation Techniques: | | | | | | | | | | | | | | |
| ActionMAE | 0.277 | 0.271 | 0.35 | 0.186 | 0.166 | 0.311 | - | - | - | - | - | - | - | - |
| DiCMoR | 0.202 | 0.465 | 0.467 | 0.152 | 0.452 | 0.454 | - | - | - | - | - | - | - | - |
| IMDer | 0.204 | 0.376 | 0.374 | 0.155 | 0.368 | 0.367 | - | - | - | - | - | - | - | - |
| Modality Reconstruction Techniques: | | | | | | | | | | | | | | |
| SURE + Gaussian MLE | 0.676 | 0.238 | 0.685 | 0.665 | 0.233 | 0.672 | 0.137 | 0.233 | 0.358 | 0.349 | 0.468 | 0.193 | 0.198 | 0.115 |
| SURE + MC DropOut | 0.653 | 0.491 | 0.669 | 0.65 | 0.466 | 0.658 | 0.243 | 0.334 | 0.174 | 0.186 | 0.41 | 0.249 | 0.222 | 0.134 |
| SURE + DeepEnsemble | 0.682 | 0.327 | 0.684 | 0.673 | 0.31 | 0.673 | 0.128 | 0.135 | 0.144 | 0.214 | 0.227 | 0.242 | 0.231 | 0.177 |
| **SURE** | 0.683 | 0.413 | 0.696 | 0.671 | 0.401 | 0.688 | 0.637 | 0.833 | 0.373 | 0.481 | 0.474 | 0.211 | 0.19 | 0.103 |

**Human Action Recognition.** As suggested in Table 3, SURE consistently delivers the best performance on downstream tasks across all scenarios. Output uncertainty most closely reflects actual error when the Watch Accel modality is available. However, we observe that a modality effective for downstream task performance may not always contribute equally to uncertainty estimation. This is likely due to the independent nature of error distributions across different modality combinations,

which leads to a divergence between downstream task performance and uncertainty estimation. Extended report with every input modalities combination is presented in Appendix A.3.1.

Table 3: Results of different approaches on UTD-MHAD Dataset.

| Model | F1 | | | | Acc | | | | Reconstruct Uncertainty Corr | | | Output Uncertainty Corr | | | | Output UCE | | | |
|---|---|---|---|---|---|---|---|---|---|---|---|---|---|---|---|---|---|---|---|
| | V(ideo) | A(ccel) | G(yro) | F(ull) | V | A | G | F | V | A | G | V | A | G | F | V | A | G | F |
| *Modality Reconstruction Techniques:* | | | | | | | | | | | | | | | | | | | |
| ActionMAE | 0.044 | 0.204 | 0.303 | 0.531 | 0.059 | 0.231 | 0.311 | 0.537 | - | - | - | - | - | - | - | - | - | - | - |
| DiCMoR | 0.069 | 0.473 | 0.52 | 0.653 | 0.033 | 0.408 | 0.472 | 0.636 | - | - | - | - | - | - | - | - | - | - | - |
| IMDer | 0.089 | 0.157 | 0.141 | 0.687 | 0.069 | 0.158 | 0.145 | 0.689 | - | - | - | - | - | - | - | - | - | - | - |
| *Uncertainty Estimation Techniques:* | | | | | | | | | | | | | | | | | | | |
| SURE + Gaussian MLE | 0.116 | 0.433 | 0.468 | 0.693 | 0.074 | 0.381 | 0.387 | 0.651 | 0.166 | 0.115 | 0.056 | 0.122 | 0.476 | 0.147 | 0.292 | 0.451 | 0.233 | 0.351 | 0.281 |
| SURE + MC DropOut | 0.156 | 0.473 | 0.595 | 0.739 | 0.09 | 0.404 | 0.571 | 0.718 | 0.122 | 0.135 | 0.171 | 0.136 | 0.486 | 0.223 | 0.512 | 0.274 | 0.149 | 0.257 | 0.137 |
| SURE + DeepEnsemble | 0.25 | 0.468 | 0.593 | 0.737 | 0.207 | 0.453 | 0.604 | 0.735 | 0.249 | 0.175 | 0.122 | 0.126 | 0.421 | 0.436 | 0.481 | 0.311 | 0.208 | 0.187 | 0.133 |
| **SURE** | 0.161 | 0.462 | 0.607 | 0.739 | 0.121 | 0.431 | 0.59 | 0.74 | 0.878 | 0.837 | 0.863 | 0.226 | 0.53 | 0.306 | 0.568 | 0.301 | 0.104 | 0.226 | 0.009 |

**Summary.** Compared to other uncertainty estimation methods, our $\mathcal{L}_{PCC}$ loss relaxes strict magnitude constraints, allowing it to efficiently learn uncertainty and accurately capture model errors based on both input and the model's stochastic nature. The recorded metrics show a strong correlation between uncertainty and error for both reconstruction and downstream tasks, further validating the effectiveness of SURE.

Table 4: Results of different SURE's variations of SURE on UTD-MHAD Dataset.

| | Model | | F1 | Acc | Reconstruct Uncertainty Corr | Output Uncertainty Corr |
|---|---|---|---|---|---|---|
| Reconstruct Ablation | (1a) | Full | 0.151 | 0.098 | - | 0.124 |
| | (1b) | Video | 0.095 | 0.094 | - | 0.128 |
| | | Accel | 0.059 | 0.081 | - | 0.322 |
| | | Gyro | 0.408 | 0.413 | - | 0.122 |
| | | Full | 0.519 | 0.525 | - | 0.524 |
| Uncertainty Est. Ablation | (2a) | Video | 0.173 | 0.117 | - | - |
| | | Accel | 0.479 | 0.427 | - | - |
| | | Gyro | 0.589 | 0.571 | - | - |
| | | Full | 0.736 | 0.727 | - | - |
| | (2b) | Video | 0.15 | 0.113 | - | 0.159 |
| | | Accel | 0.456 | 0.489 | - | 0.489 |
| | | Gyro | 0.512 | 0.462 | - | 0.237 |
| | | Full | 0.637 | 0.593 | - | 0.511 |
| Pretraining Ablation | (3) | Video | 0.031 | 0.005 | 0.684 | 0.026 |
| | | Accel | 0.226 | 0.237 | 0.675 | 0.441 |
| | | Gyro | 0.434 | 0.418 | 0.68 | 0.463 |
| | | Full | 0.615 | 0.618 | - | 0.472 |
| | **SURE** | Video | 0.161 | 0.121 | 0.878 | 0.226 |
| | | Accel | 0.462 | 0.431 | 0.837 | 0.53 |
| | | Gyro | 0.607 | 0.59 | 0.863 | 0.306 |
| | | Full | 0.739 | 0.74 | - | 0.568 |

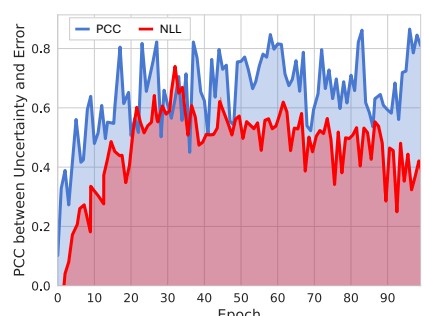

Figure 4: Correlation of estimated uncertainty with prediction error on UTD-MHAD dataset.

## 4 ANALYSES

### 4.1 ABLATION STUDY.

**Settings.** We analyze the impact of various modules on SURE's performance in both uncertainty estimation and downstream tasks. This analysis includes testing several ablated versions of SURE:

(1a) **Remove $r^i(.)$ modules**: Ignore incomplete samples during training.
(1b) **Rule-based imputation**: Replace missing modalities with zeros.
(2a) **Remove uncertainty estimation**: Train $r^i(.)$ with MSE only, no uncertainty estimation.
(2b) **Remove reconstruction uncertainty**: Train $r^i(.)$ with MSE; omit error propagation logic.
(3) **Remove pretrained weights**: Reinitialize and train backbone frameworks from scratch.

**Results.** We present the performance of all SURE variations on the UTD-MHAD dataset in Table 4. Overall, each ablation negatively impacts SURE's performance in its respective tasks. Specifically, ignoring missing modalities (1a) or using simple rule-based imputation (1b) significantly reduces downstream task performance, as a substantial portion of incomplete yet labeled data remains underutilized. Additionally, while removing uncertainty estimation logic has a negligible effect on the

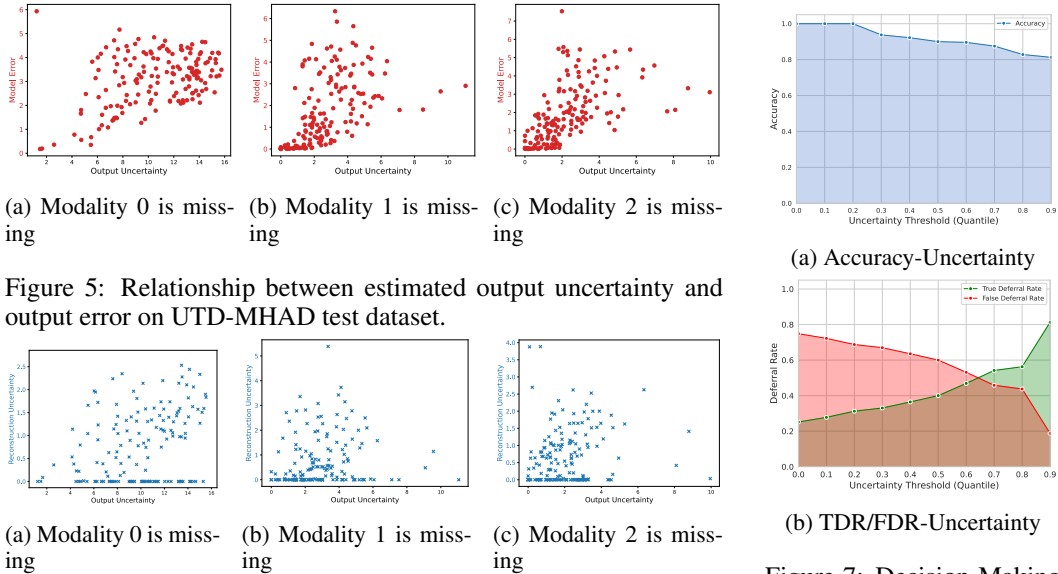

(a) Modality 0 is missing

(b) Modality 1 is missing

(c) Modality 2 is missing

Figure 5: Relationship between estimated output uncertainty and output error on UTD-MHAD test dataset.

(a) Accuracy-Uncertainty

(a) Modality 0 is missing

(b) Modality 1 is missing

(c) Modality 2 is missing

(b) TDR/FDR-Uncertainty

Figure 6: Relationship between estimated output uncertainty and reconstruction uncertainty on UTD-MHAD test dataset.

Figure 7: Decision Making Process with Uncertainty on UTD-MHAD Dataset.

final task result (2a), the inability to quantify output uncertainty from reconstructed inputs negatively impacts the accuracy of final estimations (2b). Lastly, the results from variation (3) reinforce our motivation: utilizing pretrained weights is far more efficient and beneficial, especially for smaller datasets involving similar tasks.

### 4.2 Analyses for estimated uncertainty.

**Convergence Analysis.** We visualize the correlation between estimated uncertainties and prediction errors across all training epochs in Figure 4. Compared to the Negative Log-Likelihood Loss (NLL), $\mathcal{L}_{PCC}$ demonstrates superior performance in both convergence speed and final estimation accuracy. Additionally, the shape of the NLL curve suggests instability, as the correlation trend declines after reaching its peak. Although there are fluctuations, our loss maintains an overall upward trend, eventually stabilizing in the final epochs. This experimental results are highly in accordant with our theoretical analysis of convergence points for NLL loss and our proposed loss (Appendix A.1.1).

**Reconstruction and Output Uncertainty Analysis.** To better understand the relationship between prediction error, reconstruction uncertainty, and output uncertainty, we visualize these three quantities across all test samples in the UTD-MHAD dataset, with different modalities combinations where each modality is missing. Ideally, the points should cluster along the bottom-left to top-right diagonal, indicating perfect correlation. With SURE, we observe high efficiency in estimating uncertainty for samples with large prediction errors, which aligns with its intended use as an indicator for potentially error-prone predictions (Figure 5). Notably, when output uncertainties are high, reconstruction uncertainties tend to be elevated as well (Figure 6), suggesting that uncertainties arising from the reconstruction process play a significant role in the overall uncertainty estimation. However, the visualization also indicates a tendency toward overestimating both reconstruction and output uncertainties, highlighting an area for potential improvement in future research.

### 4.3 Application: Uncertainty-informed Decision Making with SURE

**Settings.** To demonstrate the impact of SURE's uncertainty quantification on decision-making, we simulate this process using a human action recognition task with the UTD-MHAD dataset. SURE is trained with similar settings to those used in the main experiment (Table 3). After training, we use the uncertainty estimates to determine whether the model is confident enough to make a final decision or if it should defer the decision for manual inspection. Different uncertainty thresholds are set based on output uncertainty values from the test dataset. For each threshold, predictions with

uncertainty higher than the threshold are deferred, and we record **Accuracy**, **True Deferral Rate**, and **False Deferral Rate** (representing the rate of correctly and incorrectly deferred samples) across all test samples.

**Results.** Figure 7a shows that as more uncertain predictions are deferred, the remaining predictions become more challenging, resulting in a decline in accuracy. This suggests that while the deferral strategy successfully excludes uncertain predictions, it also leaves a set of samples that are inherently harder to predict accurately. Additionally, Figure 7b demonstrates that as the uncertainty threshold increases, the true deferral rate rises, while the false deferral rate falls. This indicates that the model effectively identifies uncertain predictions (leading to more true deferrals) while reducing unnecessary deferrals. The point at which the true deferral rate surpasses the false deferral rate represents an optimal balance, maximizing decision quality and minimizing unwarranted deferrals. Combining the extended decision-making process under missing modality conditions (as presented in Appendix A.3.2), this analysis indicates that SURE's estimated uncertainty is a reliable indicator for ensuring high prediction quality.

## 5 RELATED WORKS

**Multimodal missing modalities.** Recent research has focused on developing models that are resilient to missing modalities Ma et al. (2021; 2022); Poklukar et al. (2022); Woo et al. (2023); Lee et al. (2023a). For example, Smil Ma et al. (2021) employs Bayesian meta-learning to approximate latent features for modality-incomplete data. GMC Poklukar et al. (2022) maintains geometric alignment in multimodal representations, allowing unimodal representations to substitute for missing modalities. Similarly, ActionMAE, inspired by the masked autoencoder framework Feichtenhofer et al. (2022); Bachmann et al. (2022), learns to predict the latent representation of a missing modality by randomly dropping its feature token and reconstructing it. Despite success in specific scenarios, these approaches rely heavily on labeled data and lack uncertainty analysis for incomplete inputs, reducing their real-world reliability. SURE addresses these gaps by leveraging pretrained models with fewer labeling requirements and providing a reliable system for estimating uncertainty in both reconstruction and output.

**Uncertainty Estimation.** Recent methods for uncertainty estimation in predictions primarily rely on Bayesian models Lakshminarayanan et al. (2017); Kendall & Gal (2017c). However, while these models can estimate uncertainty, their predictive performance often lags behind other approaches. Some post-hoc works have explored using Laplace approximation to estimate uncertainty Daxberger et al. (2021); Eschenhagen et al. (2021), but these methods require computing the Hessian matrix, making them infeasible for high-dimensional problems Fu et al. (2018). Another direction involves test-time data augmentation Wang et al. (2019b); Ayhan & Berens (2018), where multiple outputs are perturbed to estimate uncertainty. However, this approach is sometimes poorly calibrated, which is critical for accurate uncertainty estimation Gawlikowski et al. (2023). SURE offers a more efficient alternative by estimating uncertainty without compromising predictive performance on downstream tasks. Unlike Laplace approximation, SURE avoids computational issues in high-dimensional spaces, and it does not rely on test-time perturbations, ensuring better-calibrated uncertainty estimates across diverse settings. Additionally, SURE imposes no assumptions on the output size, making it more flexible for a variety of applications.

## 6 CONCLUSION

**Contributions.** This work introduces SURE (Scalable Uncertainty and Reconstruction Estimation), which leverages pretrained multimodal frameworks for small datasets with missing modalities using latent space reconstruction. SURE integrates uncertainty estimation via a Pearson Correlation-based loss and error propagation, ensuring reliable predictions and adaptability across tasks and networks. It achieves state-of-the-art results in both downstream performance and uncertainty estimation.

**Limitations.** In developing SURE, we observed that certain modalities dominate the reconstruction process, making it easier to predict missing ones but causing significant performance drops when unavailable. This imbalance, unexplored in the current SURE framework, may limit the development of robust reconstruction modules and presents a valuable direction for future work.

## REPRODUCABILITY STATEMENT

We have made extensive efforts to ensure the reproducibility of our work, focusing on several key areas:

- Code Availability: The complete codebase for this work, including all models, training scripts, and evaluation procedures, is prepared. Upon acceptance, this code will be open-sourced and made publicly available on GitHub.
- Dataset Preparation: Detailed instructions for dataset setup, including any preprocessing steps and data splits used in our experiments, are provided in Section 3, enabling other researchers to replicate our exact experimental conditions.
- Hardware and Hyperparameters: A comprehensive description of the hyperparameters used in our experiments, including optimization settings, the GPUs used, and other configuration details, is provided in Appendix A.2.
- Architecture Transparency: Detailed descriptions of our model architectures are provided in Appendix A.1.4, ensuring others can understand and reconstruct the models accurately.
- Evaluation Metrics: The exact definitions of all evaluation metrics used are provided in Section 3 of the main paper.

By offering this comprehensive set of resources, we aim to facilitate the reproduction of our results by the research community. We believe that this level of transparency is crucial for advancing the field and supporting thorough validation and extension of our work.

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

# A Appendix

## A.1 SURE's Additional Details

### A.1.1 Negative Log Likelihood Loss for Uncertainty Estimation

**Analysis for the convergence of $\mathcal{L}_{NLL}(.,.)$.**

The detailed derivation of the gradient of $\mathcal{L}_{NLL}$ with respect to prediction $\tilde{y}_i$ is:

$$\frac{\partial \mathcal{L}_{NLL}(\tilde{y}, \tilde{\sigma}^2)}{\partial \tilde{y}_i} = \frac{\partial}{\partial \tilde{y}_i} \sum_{i=1}^{N} \frac{\tilde{\epsilon}_i^2}{2\tilde{\sigma}_i^2} + \frac{\log\left(\tilde{\sigma}_i^2\right)}{2} = \frac{\tilde{\epsilon}_i}{\tilde{\sigma}_i^2} = \frac{\tilde{y}_i - y_i}{\tilde{\sigma}_i^2}.$$

Solving $\frac{\partial \mathcal{L}_{NLL}(\tilde{y}, \tilde{\sigma}^2)}{\partial \tilde{y}_i} = 0$ give us the closed form solution $\tilde{y}_i^* = y_i$ (One can further verify sufficient condition $\frac{\partial^2 \mathcal{L}_{NLL}(\tilde{y}_i, \tilde{\sigma}_i^2)}{\partial \tilde{y}_i^2} = \frac{1}{\tilde{\sigma}_i^2} > 0$ hold true $\forall i$).

Similarly, gradient of $\mathcal{L}_{NLL}$ with respect to $\tilde{\sigma}_i^2$ is:

$$\frac{\partial \mathcal{L}_{NLL}(\tilde{y}, \tilde{\sigma}^2)}{\partial \tilde{\sigma}_i^2} = \frac{\partial}{\partial \tilde{\sigma}_i^2} \sum_{i=1}^{N} \frac{\tilde{\epsilon}_i^2}{2\tilde{\sigma}_i^2} + \frac{\log\left(\tilde{\sigma}_i^2\right)}{2} = \frac{1}{2\left(\tilde{\sigma}_i^2\right)^2}\left(\tilde{\sigma}_i^2 - \tilde{\epsilon}_i^2\right).$$

Setting $\frac{\partial \mathcal{L}_{NLL}(\tilde{y}, \tilde{\sigma}^2)}{\partial \tilde{\sigma}_i^2} = 0$ yield $\tilde{\sigma}_i^{2*} = \tilde{\epsilon}_i^2$. Verifying the sufficient condition:

$$\left. \frac{\partial^2 \mathcal{L}_{NLL}}{\partial \left(\tilde{\sigma}_i^2\right)^2}\right|_{\tilde{\sigma}_i^2 = \tilde{\epsilon}_i^2} = \frac{1}{2}\left(-\frac{1}{\left(\tilde{\epsilon}_i^2\right)^2} + \frac{2\tilde{\epsilon}_i^2}{\left(\tilde{\epsilon}_i^2\right)^3}\right) = \frac{1}{2}\left(-\frac{1}{\tilde{\epsilon}_i^4} + \frac{2}{\tilde{\epsilon}_i^4}\right) = \frac{1}{2} \cdot \frac{1}{\tilde{\epsilon}_i^4} > 0$$

This test result indicates a local minimum at $\tilde{\sigma}_i^{2*} = \tilde{\epsilon}_i^2$.

The issue optimizing $\mathcal{L}_{NLL}(\tilde{y}, \tilde{\sigma}^2)$ come up when $\tilde{\epsilon}_i \to 0$, this pull the gradient $\frac{\partial \mathcal{L}_{NLL}(\tilde{y}_i, \tilde{\sigma}_i^2)}{\partial \tilde{\sigma}_i^2}$ to the form $\frac{0}{0}$, which is mathematically undefined. This pose a significant issue for gradient-based optimization algorithms like Gradient Descent and cause arbitrary potential issues (gradient vanishing/exploding, numerical under/overflow sensitive to small changes of $\tilde{\epsilon}^2$, etc).

**Analysis for the convergence of $\mathcal{L}_{PCC}(.,.)$.**

For this analysis, we focus on the convergence for finding optimal $\tilde{\sigma}_i^{2*}$ of $\mathcal{L}_{PCC}(.,.)$. With:

$$\mathcal{L}_{PCC}(\tilde{\sigma}^2, \tilde{\epsilon}^2) = 1 - r(\tilde{\sigma}^2, \tilde{\epsilon}^2);$$

$$r(\tilde{\sigma}^2, \tilde{\epsilon}^2) = \frac{\sum_{i=1}^{N}\left(\tilde{\sigma}_i^2 - \mu_{\sigma^2}\right)\left(\tilde{\epsilon}_i^2 - \mu_{\epsilon^2}\right)}{\sqrt{\sum_{i=1}^{N}\left(\tilde{\sigma}_i^2 - \mu_{\sigma^2}\right)^2}\sqrt{\sum_{i=1}^{N}\left(\tilde{\epsilon}_i^2 - \mu_{\epsilon^2}\right)^2}} := \frac{A}{B}.$$

We have:

$$\frac{\partial \mathcal{L}_{PCC}(\tilde{y}, \tilde{\sigma}^2)}{\partial \tilde{\sigma}_i^2} = -\frac{\partial r(\tilde{y}, \tilde{\sigma}^2)}{\partial \tilde{\sigma}_i^2} = -\frac{1}{B}\frac{\partial A}{\partial \tilde{\sigma}_i^2} + \frac{A}{B^2}\frac{\partial B}{\partial \tilde{\sigma}_i^2}.$$

$$\frac{\partial A}{\partial \tilde{\sigma}_i^2} = \frac{\partial}{\partial \tilde{\sigma}_i^2} \sum_{j=1}^{N}\left(\tilde{\sigma}_j^2 - \mu_{\sigma^2}\right)\left(\tilde{\epsilon}_j^2 - \mu_{\epsilon^2}\right)$$

$$= \sum_{j=1}^{N}(\delta_{ij} - \frac{1}{N})(\tilde{\epsilon}_j^2 - \mu_{\epsilon^2}) \text{ (where } \delta_{ij} = 1 \text{ if } i = j \text{ else } 0)$$

$$= \tilde{\epsilon}_i^2 - \mu_{\epsilon^2} \text{ (Since } \frac{1}{N}\sum_{j=1}^{N}\tilde{\epsilon}_i^2 = \mu_{\epsilon^2}).$$

Also,

$$\frac{\partial B}{\partial \tilde{\sigma}_i^2} = \frac{\partial}{\partial \tilde{\sigma}_i^2} \sqrt{\sum_{j=1}^{N} \left(\tilde{\sigma}_j^2 - \mu_{\sigma^2}\right)^2} \sqrt{\sum_{j=1}^{N} \left(\tilde{\epsilon}_j^2 - \mu_{\epsilon^2}\right)^2}$$

Denoting $\sigma_{\tilde{\sigma}^2} := \sum_{j=1}^{N} \left(\tilde{\sigma}_j^2 - \mu_{\sigma^2}\right)^2$, $\sigma_{\tilde{\epsilon}^2} := \sum_{j=1}^{N} \left(\tilde{\epsilon}_j^2 - \mu_{\epsilon^2}\right)^2$, we have:

$$\frac{\partial B}{\partial \tilde{\sigma}_i^2} = \sigma_{\tilde{\epsilon}^2} \frac{1}{2\sigma_{\tilde{\sigma}^2}} \frac{\partial}{\partial \tilde{\sigma}_i^2} \sum_{j=1}^{N} \left(\tilde{\sigma}_j^2 - \mu_{\sigma^2}\right)^2$$

$$= \sigma_{\tilde{\epsilon}^2} \frac{1}{2\sigma_{\tilde{\sigma}^2}} \left[ \sum_{j=1}^{N} 2(\tilde{\sigma}_j^2 - \mu_{\sigma^2})(\delta_{ij} - \frac{1}{N}) \right]$$

$$= \sigma_{\tilde{\epsilon}^2} \frac{1}{2\sigma_{\tilde{\sigma}^2}} \left[ 2(\tilde{\sigma}_i^2 - \mu_{\sigma^2}) - \frac{2}{N} \sum_{j=1}^{N} (\tilde{\sigma}_j^2 - \mu_{\sigma^2}) \right]$$

$$= \sigma_{\tilde{\epsilon}^2} \sigma_{\tilde{\sigma}^2} \frac{\tilde{\sigma}_i^2 - \mu_{\sigma^2}}{\sigma_{\tilde{\sigma}^2}^2}$$

Assembling the results, we have:

$$\frac{\partial \mathcal{L}_{PCC}(\tilde{y}, \tilde{\sigma}^2)}{\partial \tilde{\sigma}_i^2} = -\frac{\partial r(\tilde{y}, \tilde{\sigma}^2)}{\partial \tilde{\sigma}_i^2} = -\frac{1}{B} \frac{\partial A}{\partial \tilde{\sigma}_i^2} + \frac{A}{B^2} \frac{\partial B}{\partial \tilde{\sigma}_i^2}$$

$$= -\frac{\tilde{\epsilon}_i^2 - \mu_{\epsilon^2}}{\sigma_{\tilde{\epsilon}^2} \sigma_{\tilde{\sigma}^2}} + \frac{\tilde{\sigma}_i^2 - \mu_{\sigma^2}}{\sigma_{\tilde{\sigma}^2}^2} * \frac{\sum_{j=1}^{N} \left(\tilde{\sigma}_j^2 - \mu_{\sigma^2}\right)\left(\tilde{\epsilon}_j^2 - \mu_{\epsilon^2}\right)}{\sigma_{\tilde{\epsilon}^2} \sigma_{\tilde{\sigma}^2}}$$

$$= -\frac{\tilde{\epsilon}_i^2 - \mu_{\epsilon^2}}{\sigma_{\tilde{\epsilon}^2} \sigma_{\tilde{\sigma}^2}} + \frac{\tilde{\sigma}_i^2 - \mu_{\sigma^2}}{\sigma_{\tilde{\sigma}^2}^2} * r(\tilde{\sigma}^2, \tilde{\epsilon}^2)$$

$$= \frac{1}{\sigma_{\tilde{\sigma}^2}} \left[ \frac{\tilde{\sigma}_i^2 - \mu_{\sigma^2}}{\sigma_{\tilde{\sigma}^2}} * r(\tilde{\sigma}^2, \tilde{\epsilon}^2) - \frac{\tilde{\epsilon}_i^2 - \mu_{\epsilon^2}}{\sigma_{\tilde{\epsilon}^2}} \right]$$

$$= \frac{1}{\sigma_{\tilde{\sigma}^2}} \left[ \sigma_{\tilde{\sigma}^2} * r(\tilde{\sigma}^2, \tilde{\epsilon}^2) - \sigma_{\tilde{\epsilon}^2} \right].$$

This last result suggest the gradient $\frac{\partial \mathcal{L}_{PCC}(\tilde{y}, \tilde{\sigma}^2)}{\partial \tilde{\sigma}_i^2}$ involves all standardized variables, which are within a manageable numerical range, reducing the risk of numerical instability. In addition, there is no divisions by $\tilde{\sigma}_i^2$, hence stabilize the training process even in the event when $\tilde{\epsilon}_i^2 \to 0$.

### A.1.2 RECONSTRUCTION MODULES.

SURE involves a set of reconstruction modules to best leverage the pretrained models' weights. Each reconstruction module is tailored for a specific modality, hence this reconstruction logic is linearly scale with the total number of modalities.

**Design.** While not mentioned in SURE logic, it should be noted that all $z^j$ are first linearly projected into a shared latent space wherever needed, before passing to the reconstruction modules. This step involves a single matrix multiplication done per modality, and the learnable matrix is trained together with the reconstruction modules. With that, all $r^i(.)$'s are working with the same input latent space, we unify the design of $r^i(.)$'s to be identical across different modalities. Specifically, the design of reconstruction module $r^i(.)$ is kept as simple as possible, with the major component as Fully Connected layers and ReLU activations as follow:

$$r_{share}^i(z^j) = FC(ReLU(FC(z^j))),$$
$$r_{\mu}^i(z^j) = FC(ReLU(FC(ReLU(r_{share}^i(z^j))))), \qquad (10)$$
$$r_{\sigma}^i(z^j) = SoftPlus(FC(ReLU(FC(ReLU(r_{share}^i(z^j)||r_{\mu}^i(z^j))).$$

In Equation 10, $||$ denotes the concatenation operation, and $SolfPlus()$ activation is used to ensure the positiveness of returned uncertainty.

**Complexity.** Below, we analyze the complexity of the chosen reconstruction modules. Table 5 lists hyper-parameters involved in the analysis.

Table 5: $r^i(.)$ related hyper-parameters

| Notation | Description |
|----------|-------------|
| $M$ | number of modalities |
| $L$ | number of FC layers (in total) |
| $d_i$ | hidden dimension of $i^th$ layer's output |
| $d_0$ | input dimension |

**Time Complexity.** Assume a single multiplication or summation operation can be performed in unit time ($\mathcal{O}(1)$). We have the calculation for number of operations in a forward pass as follows.

Within the $i^th$ FC layer:

$$d_{i-1} * d_i + di,$$

Over $L$ layers:

$$\sum_{i=1}^{L} d_{i-1} * d_i + di.$$

In our implementations, we choose the same dimensions for all hidden outputs (same $d = d_i \forall i = 1, \ldots, L$), and there are $M$ modules $r^i(.)$. With this, the total number of operation is:

$$M \sum_{i=1}^{L} d_{i-1} * d_i + di = M * L * d * (d+1) = \mathcal{O}(M * L * d^2)$$

By utilizing matrix product and GPU acceleration, $d^2$ operations can in fact be performed in $\mathcal{O}(1)$ time, make the whole time complexity for individual branches be $\mathcal{O}(M * L)$, which is linearly scaled with $M$.

**Space Complexity.** Regarding the space complexity, within $i^{th}$ layer, beside the need for storing parameter matrix of size $(d_{i-1} + 1) \times d_i$, output after performing $ReLU$ activation are also stored to later perform back-propagation. Hence, the total number of stored parameters is:

$$(d_{i-1} + 1) * d_i + d_i = (d_{i-1} + 2) * d_i.$$

Following similar derivation with $L$ layers and $M$ branches, replacing $d = d_i \forall i = 1, \ldots, L$, we have the total space complexity is:

$$M * L * (d + 2) * d = \mathcal{O}(M * L * d^2).$$

Despite utilizing straightforward reconstruction procedure, SURE demonstrates effective reconstruction in the latent space while maintaining an overall additional time and space complexity linearly scaled with $M$ - the number of all modalities and $L$ - the number of FC layers (6 in our implementation including both reconstruction and uncertainty heads).

### A.1.3  EXTENSION TO M MODALITIES.

For extension to $M$ modalities, we train the reconstruction module using $\mathcal{L}_{rec}$. We use each of the available modalities as the ground-truth output and the rest available modalities as input to predict. In the second phase, we freeze all of the reconstruction modules and train the classifier head with $\mathcal{L}_{downstream}$. For each sample with missing modalities, we reconstruct them with remaining available ones, and perform simple average operation to obtain the final reconstruction. Algorithm 1 summarize the whole training process of SURE for $m \geq 2$ modalities.

**Algorithm 1** SURE training process

**Input**:
▷ $\mathcal{D}_{train} = \{(\mathbf{x}_k^i); \mathbf{y}_k | i \in \mathcal{I}_k - \text{set of indices for available modalities in sample } k^{th}\}$.
▷ $f^i(.)$ - frozen pretrained projectors; $r^i(.)$ - reconstruction modules $(i = 1, \ldots, M)$.
▷ $\omega(.)$ - frozen pretrained fusion module; $g(.)$ - classifier head.
**Output**:
▷ $r^{i*}(.)$ - Trained reconstruction modules; $g^*(.)$ - Trained classifier head $(i = 1, \ldots, M)$.

1:  *Initialize $r^i(.)$'s and $g(.)$*
      ▷ Train reconstruction modules
2:  **for** mini-batch $\mathcal{B} \in \mathcal{D}_{train}$ **do**
3:    $l_{rec} \leftarrow 0$;
4:    **for** $i \in \{1, \ldots, M\}$ **do**
5:      $l_{rec}^i \leftarrow 0$;
6:      **for** $j \in \{1, \ldots, M\}; j \neq i$ **do**
7:        $\mathbf{z}_k^i = f^i(\mathbf{x}_k^i)$  $(\forall k : i \in \mathcal{I}_k)$;
8:        $\tilde{\mathbf{z}}_k^i, \tilde{\sigma}_k^i \leftarrow r^i(\mathbf{z}_k^j)$  $(\forall k : i, j \in \mathcal{I}_k)$;
9:        $l_{rec}^i \leftarrow l_{rec}^i + \mathcal{L}_{rec}(\mathbf{z}_i; \mathbf{z}_j)$;
10:      **end for**
11:      $l_{rec} \leftarrow l_{rec} + l_{rec}^i$;
12:    **end for**
13:    Backprop with $l_{rec}$;
14:    Optimizer step;
15:  **end for**

16:  Freeze reconstructed modules $r^i(.)$;
      ▷ Train classifier head
17:  **for** mini-batch $\mathcal{B} \in \mathcal{D}_{train}$ **do**
18:    $\mathbf{z}_k^i = f^i(\mathbf{x}_k^i)$  $(\forall i : i \in \mathcal{I}_k)$;
19:    For $\forall i, j; i \notin \mathcal{I}_k, j \in \mathcal{I}_k$:
20:      $\tilde{\mathbf{z}}_{j-k}^i, \tilde{\sigma}_{j-k}^i = r^i(\mathbf{x}_k^j)$;
21:      $\tilde{\mathbf{z}}_k^i = \texttt{average}(\tilde{\mathbf{z}}_{j-k}^i)$;
22:      $\tilde{\sigma}_{\tilde{z}_i}^2 = \texttt{average}(\tilde{\sigma}_{j-k}^i)$;
23:      $\tilde{\mathbf{y}}_k, \tilde{\sigma}_{\omega-k} \leftarrow g(\omega(\mathbf{z}_k^i, \tilde{\mathbf{z}}_k^j))$
24:    $\tilde{\sigma}_{input-k} \leftarrow \sum_{i \notin \mathcal{I}_k} \left(\frac{\partial \omega}{\partial \tilde{z}_k^i}\right)^2 \tilde{\sigma}_{\tilde{z}_i}^2$;
25:    $\tilde{\sigma}_{\tilde{y}_k}^2 \leftarrow \tilde{\sigma}_{input-k} + \tilde{\sigma}_{\omega-k}$;
26:    $l_{downstream} \leftarrow \mathcal{L}_{downstream}(\tilde{\mathbf{y}}_k; \mathbf{y}_k)$;
27:    $l_{y-pcc} \leftarrow \mathcal{L}_{PCC}(\tilde{\sigma}_{\tilde{y}_k}^2; l_{downstream})$;
28:    Backprop with $l_{y-pcc}$ and $l_{downstream}$;
      Optimizer step;
29:  **end for**

### A.1.4 ADDITIONAL IMPLEMENTATION DETAILS

**SURE's Implementation Details.** In SURE, we reutilize pretrained multimodal frameworks chosen for specific tasks. The only replacement is the final layers producing prediction, since the classification task might involve different number of classes, and there is an additional output head for estimation of output uncertainty.

**Sentiment Analysis.** We use MMML Wu et al. (2024) trained on the CMU-MOSEI dataset Zadeh et al. (2018) as the pretrained framework. SURE's reconstruction modules are added right after the projection modules - *Text/Audio feature networks* in original paper's language Wu et al. (2024). Their fusion network are kept intact to leverage most pretrained weights as possible. We replace the last fully connected layer - classifier with two layers - one for the final output and one for estimated output uncertainty.

**Book genre classification.** We integrate SURE with MMBT Kiela et al. (2019), a pretrained framework on the MM-IMDB dataset Arevalo et al. (2020). MMBT is a bitransformer architecture, hence we consider the all the processing before positional embedding and segment embedding as the projection logic (refer to Kiela et al. (2019) for clearer architecture details), and add our reconstruction modules are inseted after this logic. The remaining transformer logic are considered fusion modules, and kept intact.

**Human Action Recognition.** We use HAMLET framework Islam & Iqbal (2020), pretrained on the large-scale MMAct dataset Kong et al. (2019) for this task. HAMLET define their projection modules as *Unimodal Feature Encoders* Islam & Iqbal (2020). SURE's reconstruction modules are included right after these encoders, while retain their original MAT module.

**Baselines' Implementation Details.** For all baselines, we also reutilize pretrained multimodal frameworks chosen for specific tasks like the adoptation with SURE. In addition, the hidden dimension used within Reconstruction-based baselines are also modified to be the same as those SURE for fair comparison.

## A.2 Environment Settings

All implementations and experiments are performed on a single machine with the following hardware setup: a 6-core Intel Xeon CPU and two NVIDIA A100 GPUs for accelerated training.

Our codebase is primarily built using *PyTorch 2.0*, incorporating *Pytorch-AutoGrad* for deep learning model development and computations. We also use tools from *Scikit-learn*, *Pandas*, and *Matplotlib* to support various experimental functionalities. The original codebase for SURE will be released publicly upon publication.

## A.3 Additional Experiments and Analyses

### A.3.1 Extended modalities missing scenarios

In Table 6, we provide a comprehensive evaluation of different frameworks across all combinations of input modalities on the UTD-MHAD dataset. This table expands on the information presented in Table 3 in the main text. The reported reconstruction uncertainty for cases with more than one available modality is averaged over all missing modalities (e.g., given (Video + Accel) inputs, the reported reconstruction uncertainty represents the average value for Gyro reconstruction). The results show that SURE consistently delivers the best performance in most uncertainty estimation scenarios while maintaining competitive results for the downstream task, underscoring its robustness across different missing modality situations.

### A.3.2 Extend Decision Making Analysis

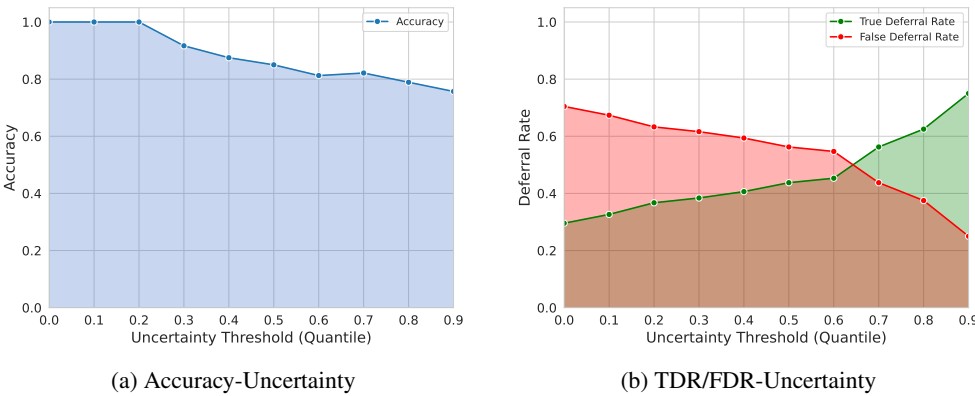

(a) Accuracy-Uncertainty                (b) TDR/FDR-Uncertainty

Figure 8: Decision Making with uncertainty when Video is missing

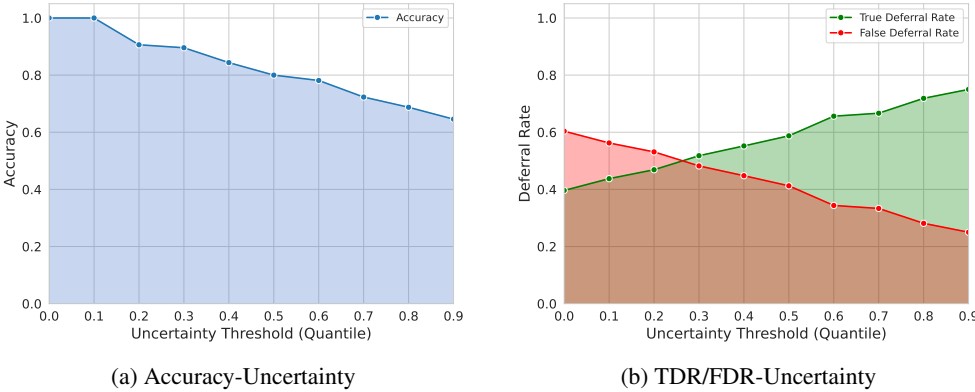

(a) Accuracy-Uncertainty                (b) TDR/FDR-Uncertainty

Figure 9: Decision Making with uncertainty when Accel is missing

Table 6: Results of different approaches on UTD-MHAD Dataset given every possible combination of input modalities.

| | Model | | F1 | Acc | Reconstruct Uncertainty Corr | Output Uncertainty Corr |
|---|---|---|---|---|---|---|
| Modal Reconstruction | ActionMAE | Video | 0.044 | 0.059 | - | - |
| | | Accel | 0.204 | 0.231 | - | - |
| | | Gyro | 0.303 | 0.311 | - | - |
| | | Video + Accel | 0.034 | 0.085 | - | - |
| | | Video + Gyro | 0.301 | 0.305 | - | - |
| | | Accel + Gyro | 0.31 | 0.306 | - | - |
| | | Full | 0.531 | 0.537 | - | - |
| | DiCMoR | Video | 0.069 | 0.033 | - | - |
| | | Watch Accel | 0.473 | 0.408 | - | - |
| | | Phone Gyro | 0.52 | 0.472 | - | - |
| | | Video + Accel | 0.524 | 0.449 | - | - |
| | | Video + Gyro | 0.536 | 0.553 | - | - |
| | | Accel + Gyro | 0.577 | 0.586 | - | - |
| | | Full | 0.653 | 0.636 | - | - |
| | IMDer | Video | 0.089 | 0.069 | - | - |
| | | Watch Accel | 0.157 | 0.158 | - | - |
| | | Phone Gyro | 0.141 | 0.145 | - | - |
| | | Video + Accel | 0.152 | 0.152 | - | - |
| | | Video + Gyro | 0.248 | 0.257 | - | - |
| | | Accel + Gyro | 0.316 | 0.278 | - | - |
| | | Full | 0.687 | 0.689 | - | - |
| Uncertainty Estimation | SURE* + Gaussian MLE | Video | 0.116 | 0.074 | 0.166 | 0.122 |
| | | Watch Accel | 0.433 | 0.381 | 0.115 | 0.476 |
| | | Phone Gyro | 0.468 | 0.387 | 0.056 | 0.147 |
| | | Video + Accel | 0.432 | 0.443 | 0.104 | 0.237 |
| | | Video + Gyro | 0.462 | 0.502 | 0.095 | 0.143 |
| | | Accel + Gyro | 0.639 | 0.67 | 0.242 | 0.29 |
| | | Full | 0.693 | 0.651 | - | 0.292 |
| | SURE* + MC DropOut | Video | 0.156 | 0.09 | 0.122 | 0.136 |
| | | Watch Accel | 0.473 | 0.404 | 0.135 | 0.486 |
| | | Phone Gyro | 0.595 | 0.571 | 0.171 | 0.223 |
| | | Video + Accel | 0.452 | 0.52 | 0.186 | 0.292 |
| | | Video + Gyro | 0.546 | 0.56 | 0.101 | 0.376 |
| | | Accel + Gyro | 0.618 | 0.639 | 0.201 | 0.417 |
| | | Full | 0.739 | 0.718 | - | 0.512 |
| | SURE* + DeepEnsemble | Video | 0.25 | 0.207 | 0.249 | 0.126 |
| | | Watch Accel | 0.468 | 0.453 | 0.175 | 0.421 |
| | | Phone Gyro | 0.593 | 0.604 | 0.122 | 0.436 |
| | | Video + Accel | 0.652 | 0.662 | 0.092 | 0.346 |
| | | Video + Gyro | 0.776 | 0.781 | 0.176 | 0.462 |
| | | Accel + Gyro | **0.839** | **0.843** | 0.278 | 0.486 |
| | | Full | 0.737 | 0.735 | - | 0.481 |
| | SURE | Video | 0.161 | 0.121 | **0.878** | 0.226 |
| | | Watch Accel | 0.462 | 0.431 | 0.837 | 0.53 |
| | | Phone Gyro | 0.607 | 0.59 | 0.863 | 0.306 |
| | | Video + Accel | 0.542 | 0.606 | 0.873 | 0.412 |
| | | Video + Gyro | 0.609 | 0.637 | 0.862 | 0.379 |
| | | Accel + Gyro | 0.679 | 0.706 | 0.455 | 0.51 |
| | | Full | 0.739 | 0.74 | - | **0.568** |

Building on the main text analysis, we simulate the decision-making process on the UTD-MHAD dataset under conditions where different modalities are missing (Figures 8, 9, 10). Each figure represents the inference scenarios when the Video, Accel, or Gyro modality is absent. Similar to the decision-making process with full modalities, incorporating uncertainty estimates in cases with missing modalities continues to guide a reliable decision-making process by adjusting different uncertainty thresholds.

### A.3.3  ADDITIONAL COMPARISON WITH PROMPT-BASED TECHNIQUES

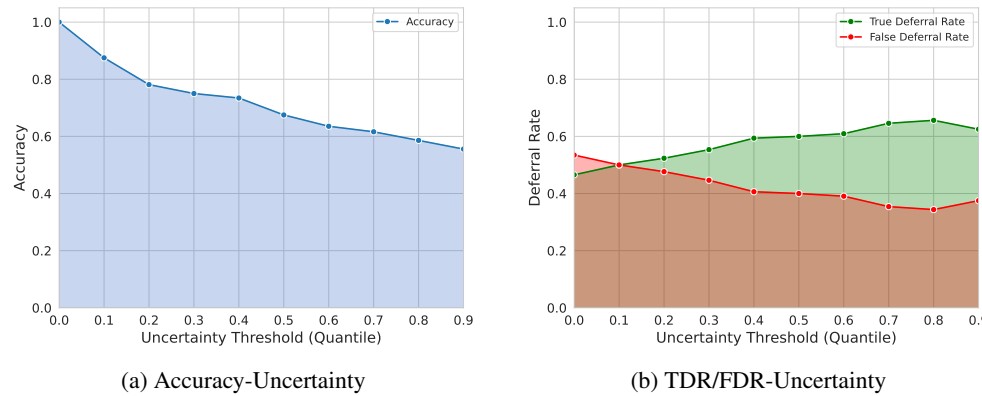

(a) Accuracy-Uncertainty  (b) TDR/FDR-Uncertainty

Figure 10: Decision Making with uncertainty when Gyro is missing

Table 7: Additional results of different approaches on CMU-MOSI Dataset.

| Model | MAE | | | Corr | | | F1 | | | Acc | | |
|---|---|---|---|---|---|---|---|---|---|---|---|---|
| | T(ext) | A(udio) | F(ull) | T | A | F | T | A | F | T | A | F |
| MPMM | 0.683 | 1.197 | 0.668 | 0.83 | 0.495 | 0.834 | 0.87 | 0.69 | 0.874 | 0.871 | 0.689 | 0.875 |
| MPLMM | 0.624 | 1.166 | 0.607 | 0.838 | 0.509 | 0.842 | 0.865 | **0.697** | 0.879 | 0.865 | **0.694** | 0.879 |
| **SURE** | **0.602** | **1.148** | **0.583** | **0.865** | **0.557** | **0.869** | **0.896** | 0.685 | **0.891** | **0.894** | 0.684 | **0.89** |

We further compare our SURE pipeline with two representative approaches that use prompt-based tuning techniques to address missing modalities Lee et al. (2023b); Guo et al. (2024). Similar to our work, these approaches also leverage pretrained multimodal pipelines for efficient training. Their key innovation lies in introducing trainable prompts to indicate the presence of missing modalities.

**Setting.** The chosen task for demonstration is Semantic Analysis task. In line with the CMU-MOSI experiment described in the main text, both frameworks are implemented using the MMML Wu et al. (2024) model, pretrained on the CMU-MOSEI dataset Zadeh et al. (2018). To ensure a fair comparison, all core modules from the original codebases of the two approaches are preserved to accurately replicate their performance. The training dataset is designed similarly to the main experiment, with 50% of modalities randomly masked and treated as missing.

**Result.** As shown in Table 7, SURE outperforms the two prompt-based approaches in handling missing modalities, achieving better downstream task performance. This advantage may stem from the limited number of learnable parameters introduced by these techniques, which likely constrain their ability to adapt effectively to scenarios with missing modalities.

