# OpenReview forum: "Are you SURE? Enhancing Multimodal Pretraining with Missing Modalities through Uncertainty Estimation"
_ICLR.cc/2025/Conference — ICLR 2025 Conference Withdrawn Submission_

### Official Review · Reviewer_dpnx · 2024-10-20

**Soundness:** 2
**Presentation:** 3
**Contribution:** 2
**Rating:** 5
**Confidence:** 4

**Summary:**

This paper introduces latent space reconstruction and uncertainty estimation to address missing modality issues. Experiments validate the effectiveness of the proposed method.

**Strengths:**

1. The proposed method achieves good results on multiple datasets.

**Weaknesses:**

1. The authors leverage a pre-trained model and only fine-tune the reconstruction module and the prediction head. This pipeline is similar to prompt-based methods [1,2]. Therefore, these baselines should be incorporated in the experiments.
2. The authors claim that existing methods overlook the unreliability of reconstruction features which may damage the quality of the final outputs. However, in [2], they also perform a reconstruction process in the latent space and design prompts to enhance the reliability of the reconstructed features.  Meanwhile, [2] also leverages a pre-trained model. Therefore, the differences should be explained to highlight your contributions.
3. Why do you choose Pearson correlation instead of other correlation coefficient? Have you tried to use other nonlinear correlation coefficient?
4. Why not train the reconstruction module and classifier together?
5. The authors should report the results of all missing cases in Table 1-3. Besides, performances of different missing ratios can be added to enhance the paper.

[1] multimodal prompting with missing modalities for visual recognition, cvpr2023

[2] multimodal prompt learning with missing modalities for sentiment analysis and emotion recognition, acl 2024

**Questions:**

See Weaknesses.

---

> ### Author Response · Authors · 2024-11-21
> **Official Response to Reviewer dpnx**
>
> Dear **Reviewer dpnx**,
>
> We sincerely thank the Reviewer for their time and thoughtful feedback on our paper. Below, we provide detailed responses to each of your questions and concerns:
>
> ### Regarding comparison with Prompt-based techniques:
> Within the limited discussion period, we attempt to put both these suggested methods [1,2] into comparison with SURE for the sentiment analysis task:
> | Model  | MAE (T)        | MAE (A)        | MAE (F)        | Corr (T)       | Corr (A)       | Corr (F)       | F1 (T)        | F1 (A)        | F1 (F)        | Acc (T)       | Acc (A)       | Acc (F)       |
> |--------|----------------|----------------|----------------|----------------|----------------|----------------|---------------|---------------|---------------|---------------|---------------|---------------|
> | MPMM [1]   | 0.683          | 1.197          | 0.668          | 0.83           | 0.495          | 0.834          | 0.87          | 0.69          | 0.874         | 0.871         | 0.689         | 0.875         |
> | MPLMM [2]  | 0.624          | 1.166          | 0.607          | 0.838          | 0.509          | 0.842          | 0.865         | **0.697**     | 0.879         | 0.865         | **0.694**     | 0.879         |
> | SURE   | **0.602**      | **1.148**      | **0.583**      | **0.865**      | **0.557**      | **0.869**      | **0.896**     | 0.685         | **0.891**     | **0.894**     | 0.684         | **0.89**      |
>
> As shown, **SURE outperforms the two prompt-based approaches in handling missing modalities, achieving better downstream task performance**. This advantage may stem from the limited number of learnable parameters introduced by these techniques, which likely constrain their ability to adapt effectively to scenarios with missing modalities.
>
> Additional comparisons with two remaining tasks are currently conducted by us, which will be included if our work is further considered.
>
> [1] Lee, Yi-Lun, et al. "Multimodal prompting with missing modalities for visual recognition." Proceedings of the IEEE/CVF Conference on Computer Vision and Pattern Recognition. 2023.
>
> [2] Guo, Zirun, et al. “Multimodal Prompt Learning with Missing Modalities for Sentiment Analysis and Emotion Recognition”. Proceedings of the 62nd Annual Meeting of the Association for Computational Linguistics, 2024.
>
> ### Regarding the differences between our work and MPLMM:
> Our work mainly diverges from this interesting study in several aspects:
> - **Scope difference**: The scope of [2] is focused on Sentimental Analysis and Emotion Recognition tasks, while **SURE does not tie in with any assumptions about tasks and types of modalities in the datasets**.
> - **Contribution difference**: [2]’s major contribution is the effective prompting technique to generate missing modality features. While they offer robust reconstructed features, they don’t explicitly quantify the reliability of these reconstructions. **Their main contribution is not contrast, but supplement our contribution**, which **offers a method to assess this reliability level beside ordinary reconstruction, and ultimately assess the final output reliability given these reconstructed inputs**.
>
> ### Regarding the evaluation metric:
> We intentionally choose PCC over other methods for following reasons:
> - **Linearity**: The enforcement of linear dependence resulting from PCC is indeed desirable. As **these relationships are expected to reflect a direct, proportional change**, Pearson’s focus on linearity is ideal for assessing whether uncertainty predictions align with the model errors.
> - **Simplicity and popularity**: With the simple linear constraint, **we theoretically show the direct relation of PCC loss with log-likelihood loss when the popular Gaussian distribution assumption is being made**. This analysis resonates the working and effectiveness of PCC loss in estimating uncertainty.
>
> ### Regarding the training process:
> We separate these two training logics to **obtain better stability**.
> While we can jointly train both the reconstruction module and the classifier, **the reconstructions during the initial training phase would be unstable and show great variances**. The use of such noisy and unstable reconstructions might even hurt or slow down the classifier learning process, and make it more challenging to assess the uncertainty of final outputs.

---

> ### Author Response · Authors · 2024-11-21
> **Official Response to Reviewer dpnx [2]**
>
> ### Regarding the table presentations:
> We want to thank you for your constructive feedback about our table presentation.
> In Table 1 and Table 2, we work with datasets involving only 2 modalities, hence **these two tables already cover every possible combination of input modalities**.
> For table 3, which involves 3 modalities, **the extended result of this experiment is provided in section A.3.1 owing to space constraint**.
>
> In our current settings, we set the missing ratio of the training dataset as $50\\%$, while there are no fixed constraints for testing the dataset. We are working on additional experiments covering other configs with different ratios during training for better empirical results. We will include those results for the camera-ready version.
>
> Thank you once again for taking the time to review our paper. If you have any additional questions, please feel free to reach out to us. We kindly ask you to consider revisiting your evaluation, and we would greatly appreciate your reconsideration to improve the score.

---

> > ### Author Response · Authors · 2024-11-28
> > **Reply to Reviewer dpnx**
> >
> > Dear Reviewer dpnx,
> >
> > Thank you once again for your effort in reviewing our paper.
> >
> > During the Discussion period, we have addressed your concerns and questions by:
> > - Adding prompt-based techniques as baselines and comparing them with SURE for the Semantic Analysis task.
> > - Highlighting our unique contributions and novelty compared to the existing work, MPLMM.
> > - Clarifying our choices for evaluation metrics, training processes, and table presentations.
> >
> > We kindly invite you to review our responses and make the most of this discussion period. Should you have any further questions, we would be happy to engage in additional discussions.

---

> ### Author Response · Authors · 2024-11-30
> **Reply to Reviewer dpnx**
>
> Dear Reviewer dpnx,
>
> Thank you for your time and effort in reviewing our paper.
>
> We have carefully addressed your concerns and questions throughout this discussion period. As we approach the end of this phase, we kindly ask if you could confirm your evaluation or share any additional feedback you might have.
>
> We greatly value your input and appreciate your thoughtful review.
>
> Best regards,
>
> The Authors

---

### Official Review · Reviewer_DTc9 · 2024-11-02

**Soundness:** 1
**Presentation:** 3
**Contribution:** 2
**Rating:** 5
**Confidence:** 4

**Summary:**

This paper focuses on an important challenge in multimodal learning—handling missing modalities, which is common in practical applications. The proposed SURE framework (Scalable Uncertainty and Reconstruction Estimation) aims to enhance pretrained multimodal models by incorporating latent-space reconstruction and uncertainty estimation. The inclusion of uncertainty estimates for both reconstruction and final predictions is valuable, as it provides insights into the model's confidence, especially when modalities are incomplete.

However, while the paper presents extensive experiments, its motivation and novelty are somewhat ambiguous. Moreover, claims regarding “scalability”, “distribution-free” and other aspects lack sufficient justification.

**Strengths:**

1. The paper brings up an important issue in multimodal learning—handling missing modalities, which is common in real-world applications where all data sources are not consistently available.
2. The framework provides both reconstruction and output uncertainty estimates, which help assess the reliability of predictions.
3. The paper includes extensive experiments across diverse tasks (sentiment analysis, genre classification, and action recognition) and datasets.

**Weaknesses:**

1. The motivation behind the paper is ambiguous. Although the introduction discusses the benefits of adapting pretrained multimodal models for downstream tasks, the rest of this paper primarily centers on uncertainty estimation. The authors could consider aligning the main content more closely with the stated motivation to improve coherence.
2. Figure 1 suggests that pretraining alone may be sufficient for handling incomplete inputs. Therefore, it remains unclear how missing modalities or incomplete inputs impact the adaptation of pretrained multimodal models.
3. The novelty of this paper may be limited, given that latent-space reconstruction and Bayesian uncertainty estimation are well-established techniques [1][2][3]. The authors should elaborate on how these approaches uniquely enhance the adaptation of pretrained multimodal models for downstream tasks, as their current combination appears somewhat trivial.
4. The authors describe the proposed Pearson Correlation Coefficient (PCC) Loss as a 'distribution-free' loss function. However, the notation in equation (4) suggests that the PCC loss still relies on the Gaussian assumption. Additionally, the uncertainty constraint for PCC loss also appears to be derived from this Gaussian assumption. The authors should further clarify the “distribution-free” property of the proposed PCC loss.

[1] Abdar, Moloud, et al. "A review of uncertainty quantification in deep learning: Techniques, applications and challenges." Information fusion 76 (2021): 243-297.

[2] Lian, Zheng, et al. "GCNet: Graph completion network for incomplete multimodal learning in conversation." IEEE Transactions on pattern analysis and machine intelligence 45.7 (2023): 8419-8432.

[3] Ma, Mengmeng, et al. "Smil: Multimodal learning with severely missing modality." Proceedings of the AAAI Conference on Artificial Intelligence. Vol. 35. No. 3. 2021.

**Questions:**

1. What does MMML stand for?
2. According to Table 3 and 3, it seems that pretraining contributes to the majority of the performance gain of SURE. For example, the performance of the proposed SURE without pretrained weights in Table 4 seems to be worse than the performance of previous methods in Table 3. However, with pretrained weights, the preformance is significantly increased and surpasses previous methods.
3. The integration of latent reconstructors and multiple uncertainty estimations could add computational overhead. A quantitative analysis of the computational costs compared to other methods would help in understanding SURE’s efficiency and scalability.
4. The authors claim that the proposed approach is scalable (**Scalable** Uncertainty and Reconstruction Estimation). However, no justification or experimental evidence is provided to support this claim.

---

> ### Author Response · Authors · 2024-11-21
> **Official Response to Reviewer DTc9**
>
> Dear **Reviewer DTc9**,
>
> We appreciate the Reviewer’s time and effort in evaluating our paper and providing insightful feedback. Below, we address your questions and concerns point by point:
>
> ### Regarding the motivation of the paper:
> The main rationale of our work can be summarized in a single sentence:
> **SURE leverages the strength of pretrained multimodal frameworks on small, incomplete datasets (with missing modalities) for similar tasks, while quantifying prediction uncertainty on these non-ideal samples.**
>
> Considering your constructive feedback, we have made several modifications to our Introduction section. These changes better express our motivation in both leveraging pretrained models, as well as assessing models’ reliability when working in such non-ideal scenarios.
>
> ### Regarding our Figure 1:
> We want to clarify that our old Figure 1 in fact highlights the observation that **pretrained models converge better and hence achieve better results compared to vanilla models, under the same small-scaled datasets**. In this case, both the pretrained and vanilla models **ignore the samples with missing modalities in the training dataset**.
>
> ### Regarding the novelty:
> We highlight here our unique novelty with this work:
> - A **plug-in solution to tackle small-scale and incomplete multimodal datasets**, this module is training-efficient but yields competitive results and well estimated uncertainty of those results.
> - A **distribution-free uncertainty estimation technique**, which requires no bias about the underlying distribution of prediction
> - A unique and intuitive approach to **apply Error Propagation in deep learning architectures**, effectively separating 2 sources of uncertainty contributing to prediction error: from the model itself and from the reconstructed input.
>
> Modality reconstruction - as suggested, are well-established, we only **utilize this strategy to meet our goal of utilizing pretrained frameworks to their best extent**.
>
> ### Regarding the distribution assumption:
> We want to clarify that the intention of Equation 4 is a demonstration showing **the relevance of minimizing PCC loss between uncertainty and error and minimizing the MSE between standardized versions of these two variables (zero mean and unit variance)**, but **not tie our method with the Gaussian assumption**.
>
> Essentially, this relevance shows that **our method in fact enforces linear dependence between the two variables after being standardized**, hence does not depend on the scale or distribution of these variables.
> We highlight this point in the Sect. 2.3.
>
> ### Regarding the use of MMML:
> For the introductory presentation, we want to demonstrate **the benefit of using pretrained models for smaller datasets of different scales**. **MMML (Multimodal multi-loss fusion network)** [1], is a state-of-the-art fusion architecture for Semantic Analysis on the CMU-MOSI dataset.
>
> [1] Wu, Zehui, et al. "Multimodal multi-loss fusion network for sentiment analysis." Proceedings of the 2024 Conference of the North American Chapter of the Association for Computational Linguistics: Human Language Technologies (Volume 1: Long Papers). 2024.
>
> ### Regarding to our main source of improvement:
> The fact that **pretraining contributes to the majority of the performance gain of SURE is not opposed but in companion with our original motivation and methodology**.
>
> **SURE is designed so that pretrained models can be used to show their effectiveness with small and corrupted datasets**, which normally are big challenges to vanilla fusion architectures (as shown in our experiments as you suggested).
>
> SURE’s **reconstruction modules are kept simple and not the main focus of our work**, and should be thought of conceptually instead, which can be replaced and strengthened.
> From Table 4, with just a simple reconstruction modules' design (mentioned in Appendix A.1.2), SURE with vanilla fusion framework still shows improved performance, as **it still enables effective utilization of every sample (even corrupted) in the training dataset**.

---

> > ### Author Response · Authors · 2024-11-21
> > **Official Response to Reviewer DTc9 [2]**
> >
> > ### Regarding the scalability and efficiency of our framework:
> > The scalability we emphasize with SURE lies in **(1) its efficient training logic by incorporating pretrained models**, and **(2) the simple choices of architectures and designs**.
> >
> > The **major overhead of our pipeline lies in the latent reconstructors**, which are **specifically tied to the architecture design of this module**. With our simple choice of architecture (stacked of FC layers with ReLU activation - appendix A.1.2), the overhead in both time and space can be theoretically shown to be $\mathcal{O}(M*L)$, with $M$ being the total number of modalities in the datasets and $L$ is the number of layers used (assuming single operation takes $\mathcal{O}(1)$). We further provide this analysis in our Appendix A.1.2 based on this choice of architecture, ensuring the Scalability of the method.
> >
> > Regarding uncertainty estimation logic, the uncertainty for reconstructed results is tied to reconstructors. For the output uncertainty caused by reconstructed input, it is effectively calculated using the Error Propagation formula (Eq. 7), which **add up no additional calculation and space to the complexity bound**, as **all quantities are available from a single backward pass** with $\mathcal{L}_{downstream}$. For uncertainty calculation in the final classifier head, it is simply designed as a linear projection layer with parameter-free activation, which **theoretically adds $\mathcal{O}(1)$ to the overall complexity**.
> >
> > To sum up, **SURE adds the total overhead of $\mathcal{O}(M*L)$ to existing fusion architectures of choice**. This overhead is **highly efficient and scalable to a great number of modalities**. In addition, it should be noted that **SURE requires no training for pretrained models**, which also yields a much more efficient training logic, compared with other methods.
> >
> > Thank you once again for reviewing our paper. We have addressed your concerns to the best of our ability in this period. Should you have any further questions, please feel free to reach out. We kindly request your reconsideration to improve the score, and we greatly appreciate your time and effort.

---

> > > ### Comment · Reviewer_DTc9 · 2024-11-24
> > >
> > > Thank the authors for their detailed response, which addressed some of my concerns. However, the overall contribution of the paper remains unclear. The authors state that the primary motivations for this work are: (1) adapting pretrained multimodal models to downstream tasks with potential missing modality issues, and (2) introducing uncertainty estimation to address these challenges. While the methodology section places significant emphasis on uncertainty estimation, which is indeed interesting, the primary performance gains seem to stem from the use of pretrained weights combined with latent modality reconstruction—a well-explored approach in prior research.
> > >
> > > In summary, while I appreciate the authors' efforts and their response, I will maintain my original rating.

---

> ### Author Response · Authors · 2024-11-28
> **Reply to Reviewer DTc9**
>
> Dear Reviewer DTc9,
>
> Thank you for the response. We further highlight our points as follow.
>
> We’d first like to emphasize that, as the Reviewer noted, our work focuses on making the pretrained multimodal models significantly “*practical*” for downstream tasks that often contain missing modalities. However, naively estimating uncertainty, e.g., using the previous approaches, without co-optimizing **the uncertainty objective (the Pearson Correlation objective) stemming from both model nature and missing modalities** with **the downstream task objective**, would not yield satisfactory performance, and therefore **we cannot take full advantages of pretrained multimodal models in practice**. For example, post-hoc techniques like Monte Carlo Dropout or DeepEnsemble do not work well across 3 different tasks under our extensive investigation (Section 3.3), compared to ours with co-optimization of these two objectives.
>
> More specifically, **to tackle this important challenge**, our work has provided *solid analysis* of this phenomenal (via our extensive experiments across tasks) and devised a *novel objective function* (with well-supported theoretical analysis), as also mentioned by the other reviewers (**Reviewers Z2fY, 5FYA**). These are the significant contributions of our work toward **maximally ensuring the practicality of the pretrained multimodal models** in real-world applications, such as in the healthcare domain or self-driving vehicles that usually have missing modalities due to external non-ideal factors during deployment.
>
> Note that, latent modality reconstruction is simply a means for us to best leverage pretrained models and estimate the uncertainty, but is not our main contribution.
>
> For these reasons, we believe that our work is unique compared to all existing approaches, as also pointed out by Reviewers **Z2fY,  5FYA**.

---

> > ### Author Response · Authors · 2024-11-30
> > **Reply to Reviewer DTc9**
> >
> > Dear Reviewer DTc9,
> >
> > Thank you for your thoughtful review and active engagement during the Discussion period.
> >
> > We have provided detailed responses addressing your comment on our contributions. **We believe our work addresses a unique yet critical aspect of the missing modality problem with a novel approach that sets it apart from existing methods**.
> >
> > As the discussion phase comes to an end, we kindly ask if you could confirm your evaluation or share any additional feedback.
> >
> > Thank you once again for your valuable insights.
> >
> > Regards,
> >
> > The Authors

---

### Official Review · Reviewer_5FYA · 2024-11-02

**Soundness:** 2
**Presentation:** 3
**Contribution:** 2
**Rating:** 5
**Confidence:** 4

**Summary:**

This paper introduces SURE (Scalable Uncertainty and Reconstruction Estimation), a framework designed to improve pretrained multimodal models when modalities are missing during training or evaluation. SURE focuses on two key areas: reconstructing missing modalities within the latent space and quantifying uncertainty for both these reconstructions and subsequent predictions. A novel Pearson Correlation Coefficient loss function and statistical error propagation are used to estimate uncertainties, improving the reliability of the reconstructed inputs and final predictions. SURE is architecture-agnostic and can adapt to various pretrained multimodal models, achieving robust and interpretable results in experiments on sentiment analysis, genre classification, and action recognition.

Overall, this paper introduces an interesting perspective for enhancing pretrained multimodal models when modalities are missing. The framework integrates techniques for estimating uncertainty and is adaptable across various architectures. However, it requires clearer theoretical justification, especially regarding how uncertainty estimation contributes to improved predictive performance. Additionally, inconsistencies in benchmark results and limited evaluation of uncertainty estimation indicate areas that need refinement. Addressing these issues would enhance both the impact and comprehensibility of the framework.

**Strengths:**

1. The proposed approach introduces uncertainty estimation to adapt pretrained multimodal models for downstream tasks with missing modalities. This method not only enhances overall performance on downstream tasks but also enables the estimation and interpretation of uncertainty arising from both the model's inherent stochasticity and incomplete inputs.

2. The proposed approach is validated across multiple benchmarks and compared against previous methods.

3. The approach is designed to be compatible with a wide range of pretrained multimodal models, making it adaptable and applicable across diverse architectures and tasks.

**Weaknesses:**

1. It is unclear how and why uncertainty estimation enhances overall performance. While Section 2.3 suggests that incorporating Bayesian learning can help model the aleatoric uncertainty, there is a lack of theoretical explanation for how this contributes to improvements in predictive performance, such as accuracy.

2. In section 3.2, the authors mentioned that the Gaussian assumption allows for a closed-form solution to uncertainty estimation, suggesting that a strong alignment or correlation between uncertainty and prediction error leads to better performance. However, if the underlying distribution does not follow the Gaussian assumption, is this correlation still necessary for achieving an optimal solution? If not, optimizing the PCC loss to align uncertainty with prediction error may need further justification.

3. The performance of IMDer on the CMU-MOSI dataset seems to be inconsistent to the results reported in the original paper. For example, the F1 score for full inputs reported in the original IMDer paper is 85.1 v.s. 62.0 reported in this paper. The authors may need to clarify the difference.

4. The evaluation of uncertainty estimation in the paper is limited. Currently, the paper assesses uncertainty estimation by calculating the correlation between uncertainty and prediction error. However, it does not intuitively explain why a high correlation between these metrics is desirable or how it contributes to the model’s reliability.

**Questions:**

1. A detailed caption is missing for Figure 3, making it difficult to understand the symbols and the flow of the proposed pipeline since their descriptions are spread across sections 2.2 to 2.4.

2. How does the number of reconstructors \( r_i \) scale with the number of modalities? Figure 3 suggests that a large number of reconstructors may be necessary to handle all possible missing-modality scenarios, which could affect the model’s scalability. A detailed explanation would clarify how the framework manages this growth efficiently.

---

> ### Author Response · Authors · 2024-11-21
> **Official Response to Reviewer 5FYA**
>
> Dear **Reviewer 5FYA**,
>
> We thank the Reviewer for spending effort to review our paper and provide valuable feedback. We address your questions and concerns point-by-point below:
>
> ### Regarding the uncertainty estimation:
> We apologize if the previous writing leads to a misunderstanding.
> What we intentionally want to convey is that **our reconstruction logic is the main factor in boosting performance** - as it **better uses training data and bridges the gap of missing modalities**, while uncertainty estimation helps **assess the reliability of the result**. We correct the Abstract and the Introduction to further clarify this point: Figure 2 is added, which illustrates the interrelation between reconstruction uncertainty, output uncertainty and the final downstream task performance.
>
> Additionally, we illustrate **the use of estimated uncertainty to produce better decision-making when deployment** by selecting a suitable uncertainty threshold, we can effectively filter out erroneous predictions - Sec 4.3.
>
> ### Regarding the assumption about the distribution:
> We want to clarify these points as follows:
> - Our method is a distribution-free technique, essentially meaning we **don’t introduce any bias regarding the underlying distribution of estimated output**. Our constraint enforces uncertainty to ideally reflect the prediction error via linear dependence regardless of the underlying distribution.
> - The Gaussian distribution analysis is **the way we connect PCC loss with a common assumption** where **our loss leads to a solution that is equivalent to the Gaussian solution with more relaxed constraints**. However, the closed-form solution of Gaussian assumption cases may not give us the right answer when the underlying distribution is not Gaussian (eg: heavy-tailed distribution).
>
> ### Regarding the baseline model (IMDer)’s results:
> These differences stem from the following differences in the training/evaluation setting:
> - **Model setup**: The original architecture shipped with IMDer is replaced with **pretrained MMML model (for the CMU-MOSI task)** to ensure similar setup across all baseline methods.
> - **Training data**: 50% sample points of each modality in the training dataset are randomly masked out and considered as missing, compared to the use of full modality samples as in IMDer. **The main decrease in performance might be attributed to this reason**.
> - **Evaluation data**: Our evaluation involves simulations for different missing modality combinations. We **mask out specific modalities over the entire test set and assess the model performance given that corrupted data**.
>
> ### Regarding the evaluation metric:
> We intentionally chose the correlation as the evaluation metric due to its ability to capture the linear dependency of these two variables which **directly reflects the desirable characteristic of uncertainty**. This choice is **in alignment with previous series of work in uncertainty estimation literature** (e.g. [1]).
>
> For a more general assessment of the uncertainty values produced by SURE, we **additionally adopt metrics UCE (Uncertainty Calibration Error)** from papers [2] for the three tasks being investigated.
>
> Summary of uncertainty assessment with UCE:
> | Model                     | CMU-MOSI (T)   | CMU-MOSI (A)   | CMU-MOSI (F)   | Book (T)      | Book (I)     | Book (F)      | UTD-MHAD (V)   | UTD-MHAD (A)   | UTD-MHAD (G)   | UTD-MHAD (F)   |
> |---------------------------|----------------|----------------|----------------|---------------|--------------|---------------|----------------|----------------|----------------|----------------|
> | SURE + Gaussian MLE       | 0.425          | 0.476          | 0.385          | **0.193**     | 0.198        | 0.115         | 0.451          | 0.233          | 0.351          | 0.281          |
> | SURE + Monte Carlo DropOut| 0.496          | 0.51           | 0.396          | 0.249         | 0.222        | 0.134         | **0.274**      | 0.149          | 0.257          | 0.137          |
> | SURE + DeepEnsemble       | 0.497          | 0.492          | 0.389          | 0.242         | 0.231        | 0.177         | 0.311          | 0.208          | **0.187**      | 0.133          |
> | SURE                      | **0.315**      | **0.429**      | **0.285**      | 0.211         | **0.19**     | **0.103**     | 0.301          | **0.104**      | 0.226          | **0.009**      |
>
> This result suggests that SURE produces good estimation of uncertainty, consistently outperforming representative baselines over the investigated tasks.
>
> [1] Upadhyay, Uddeshya, et al. "Bayescap: Bayesian identity cap for calibrated uncertainty in frozen neural networks." European Conference on Computer Vision. Cham: Springer Nature Switzerland, 2022.
>
> [2] Guo, Chuan, et al. "On calibration of modern neural networks." International conference on machine learning. PMLR, 2017.

---

> > ### Author Response · Authors · 2024-11-21
> > **Official Response to Reviewer 5FYA [2]**
> >
> > ### Regarding our caption:
> >
> > Thanks for this insightful comment about the caption of Figure 3.
> > We modify the caption of the overview figure for a quick grasp of our method, as follows:
> >
> > “Overview of proposed module. It incorporates a set of reconstruction modules $r^i(.)$’ in the middle layers of arbitrary pretrained multimodal fusion frameworks, after their latent space projection layers $f^i(.)$ and before their fusion layers $\omega(.)$. These modules consume other modality latent representation, yield reconstructed representation for the corresponding modality, together with reconstruction uncertainty. The reconstructed output replaces the role of the original missing modality, while the uncertainty is propagated through the rest of the network to capture output uncertainty caused by the reconstruction. The final classifier additionally involves the calculation of output uncertainty caused by the model's inherent nature.”
> >
> > ### Regarding the scalability of reconstructors:
> > We want to clarify that the number of reconstructors is **linearly scaled with the number of total modalities**.
> > Specifically, $r^i(.)$ is the module dedicated to reconstruct the i-th modality, which can consume any other modality representation (e.g. z^j for j != i).
> > Within the discussion period, we provide a dedicated section in Appendix A.1.2. discussing our choice of reconstruction architectures, and analysis for their complexity. We believe this attempt provides better intuition regarding the working of r^i()’s.
> >
> > Thank you again for reviewing our paper. We have addressed your concerns to the best of our ability in this rebuttal. If you have any additional questions, please do not hesitate to reach out. We kindly ask you to consider revising your score, and we sincerely appreciate your thoughtful evaluation.

---

> > > ### Author Response · Authors · 2024-11-28
> > > **Reply to Reviewer 5FYA**
> > >
> > > Dear Reviewer 5FYA,
> > >
> > > Thank you once again for your effort in reviewing our paper.
> > >
> > > During the Discussion period, we have addressed your concerns and questions by:
> > > - Clarifying the primary use of uncertainty estimation as a tool for assessing the reliability of results.
> > > - Confirming that our approach makes no assumptions about the underlying distribution of predictions.
> > > - Explaining the key sources of performance differences in one of our baselines, primarily due to variations in experimental settings.
> > > - Conducting additional experiments to evaluate uncertainty quality using the UCE (Uncertainty Calibration Error) metric.
> > >
> > > We kindly invite you to review our responses and make the most of this discussion period. If you have further questions, we are happy to continue the discussion.

---

> ### Author Response · Authors · 2024-11-30
> **Reply to Reviewer 5FYA**
>
> Dear Reviewer 5FYA,
>
> We deeply appreciate the time and effort you have devoted to reviewing our paper.
>
> During this discussion period, we have carefully addressed your concerns and questions to the best of our ability. As the discussion phase nears its conclusion, we kindly ask if you could confirm your evaluation or provide any additional feedback.
>
> Thank you again for your thoughtful review and valuable input.
>
> Best regards,
>
> The Authors

---

### Official Review · Reviewer_Z2fY · 2024-11-03

**Soundness:** 3
**Presentation:** 3
**Contribution:** 3
**Rating:** 5
**Confidence:** 4

**Summary:**

This paper presents several key contributions through the development of the SURE (Scalable Uncertainty and Reconstruction Estimation) framework. First, SURE introduces a novel approach for handling missing modalities by incorporating latent space reconstruction
techniques into pretrained multimodal models. Second, SURE addresses the critical challenge of evaluating the reliability of both reconstructed inputs and final outputs by integrating uncertainty estimation. This is achieved through (1) a Pearson Correlation-based loss function and (2) the first application of error propagation in deep network training, allowing SURE to effectively quantify
uncertainties from multiple sources. Third, SURE is adaptable to a wide range of pretrained multimodal networks, demonstrating its robustness across various datasets and tasks, ultimately enhancing both interpretability and performance in multimodal learning scenarios.

**Strengths:**

* The theoretical proofs and methodology of this article are solid and valid.

* The Error Propagation and Uncertainty Estimation proposed by the authors is interesting and unique.

* The authors implemented diverse experiments on multimodal datasets from different domains.

**Weaknesses:**

* A great deal of multimodal works [1,2,3,4,5] related to modal absence has been neglected, leading to results and limitations that are difficult to measure.
These multimodal efforts are the main current SOTA in the field of studying modal absence across different modeling paradigms, including learning robust joint representations and reconstructions. It would be useful for the authors to clarify and compare the differences with these works.

* The writing of the article is not good enough. The author does not discuss and explain in depth many of the concepts and structures introduced. For example, there is no clear explanation of what exactly constitutes the reconstruction module described in the methods section.

* The authors consider a limited number of modal absence scenarios, and the study of modal absence should encompass both intramodal absence and intermodal absence obeying real-world scenarios. Then, the authors seem to follow a specific setup, leading to a lack of generalization of the experimental results.

[1] Li, Mingcheng, et al. "Correlation-Decoupled Knowledge Distillation for Multimodal Sentiment Analysis with Incomplete Modalities." Proceedings of the IEEE/CVF Conference on Computer Vision and Pattern Recognition. 2024.

[2] Mengmeng Ma, Jian Ren, Long Zhao, Sergey Tulyakov, Cathy Wu, and Xi Peng. Smil: Multimodal learning with severely missing modality. In AAAI.

[3] Zheng Lian, Lan Chen, Licai Sun, Bin Liu, and Jianhua Tao. Gcnet: graph completion network for incomplete multimodal learning in conversation. IEEE Transactions on Pattern Analysis and Machine Intelligence, 2023.

[4]Hai Pham, Paul Pu Liang, Thomas Manzini, Louis-Philippe Morency, and Barnabas P ´ oczos. Found in translation: Learn- ´
ing robust joint representations by cyclic translations between modalities. In Proceedings of the AAAI Conference on Artificial Intelligence (AAAI), pages 6892–6899, 2019.

[5] Li, Mingcheng, et al. "A Unified Self-Distillation Framework for Multimodal Sentiment Analysis with Uncertain Missing Modalities." Proceedings of the AAAI Conference on Artificial Intelligence. Vol. 38. No. 9. 2024.

**Questions:**

* Why use MMML specifically for the introductory introductory presentation? The reader cannot understand where the representation is. Please explain the MMML framework in detail. Would it still be similar if it was replaced with another model?

---

> ### Author Response · Authors · 2024-11-21
> **Official Response to Reviewer Z2fY**
>
> Dear **Reviewer Z2fY**,
>
> We appreciate you taking the time to review our paper and provide valuable feedback. Please find detailed responses to your questions and concerns below.
>
> ### Regarding the choice of baselines:
> We deliberately select methods that **address missing modalities through reconstruction** as our baselines. This approach is suitable to incorporate into pretrained multimodal pipelines, as it **requires no architecture/model change**, hence effectively leveraging the strengths of the pretrained models - which align with our primary motivation. Additionally, using these baselines enables a **more meaningful comparison**, as the same pretrained pipelines can be applied uniformly across all methods. The chosen baselines are also current SOTAs in reconstruction-based techniques, covering both **raw data reconstruction** (IMDer, DiCMoR - 2024) and **latent reconstruction** (ActionMAE - 2023), which are two main directions of this approach.
>
> Within this limited discussion period, we additionally **compare SURE with two recent works applying Prompt-based techniques** [1,2], which are also designed to leverage pretrained models and tackle missing modalities - Thanks to **Reviewer dpnx’s** suggestions. In summary, the results for Semantic Analysis task suggests slightly better performance of SURE over these methods, highlighting our efficiency in dealing with missing modalities. We kindly refer the Reviewer to our response to Reviewer dpnx, or to our revised Appendix A.3.3 for additional details.
>
> [1] Lee, Yi-Lun, et al. "Multimodal prompting with missing modalities for visual recognition." Proceedings of the IEEE/CVF Conference on Computer Vision and Pattern Recognition. 2023.
>
> [2] Guo, Zirun, et al. “Multimodal Prompt Learning with Missing Modalities for Sentiment Analysis and Emotion Recognition”. Proceedings of the 62nd Annual Meeting of the Association for Computational Linguistics, 2024.
>
> ### Regarding our writings:
> We have made following major changes in this rebuttal phase to better explain our ideas and concepts:
> - **Introduction**: We additionally modify to better introduce the context and problem: Tacking small datasets with missing modality by leveraging pretrained networks, assess their reliability via uncertainty estimation.
> - **Method**: we provide additional reference to Appendix, where details about different modules and architecture choices are well defined and analyzed (including reconstruction modules).
> - **Evaluation**: we provide reference to Appendix, where details about the integration of SURE with pretrained frameworks are discussed. Text modifications regarding additional experiments are also made.
>
> We hope these changes satisfactorily explain our method. We are happy to further discuss and clarify if needed.
>
> ### Regarding the choice of modal absence scenarios:
> There are several reasons for us to tackle intermodal missing:
> - We follow **the majority of literature dealing with multimodal missing**, which all focus in intermodal absence settings.
> - Intramodal absence is in fact **missing data in unimodal settings** - there are already many different lines of work extensively investigating this problem: data imputation, reconstruction, …
>
> That being said, while sticking to intermodal absence, we already target the most generic experiment setting that covers most real-world scenarios:
> - Missing modalities can presented in **both training or evaluation phases**.
> - Different input combinations, essentially **cover every combination of missing modalities** (When only a single modality is presented - Main text experiments, every possible combinations - Appendix A.3.1.)
>
> We’re open to consider running any additional experiments the Reviewer suggests to better improve our paper.
>
> ### Regarding the choice of MMML:
> The use of MMML is just for **demonstration purposes**. MMML [3] is a state-of-the-art fusion architecture for Semantic Analysis on the CMU-MOSI dataset. The main goal of this toy experiment is to show **the benefit of using pretrained models for smaller datasets of different scales**.
>
> [3] Wu, Zehui, et al. "Multimodal multi-loss fusion network for sentiment analysis." Proceedings of the 2024 Conference of the North American Chapter of the Association for Computational Linguistics: Human Language Technologies (Volume 1: Long Papers). 2024.
>
> Once again, thank you for reviewing the paper. We think we have solved your problem as much as possible in rebuttal, If you have any further questions, please do not hesitate to contact us. If possible, we would like to thank you for reconsidering to improve your score.

---

> > ### Author Response · Authors · 2024-11-28
> > **Reply to Reviewer Z2fY**
> >
> > Dear Reviewer Z2fY,
> >
> > Thank you again for your effort reviewing our paper.
> >
> > During the Discussion period, we have addressed your concerns and questions by:
> > - Clarifying the reasons for our choice of baselines.
> > - Making major revision in our manuscript by putting constructive feedbacks of you and other Reviewers into account.
> > - Clarifying the reasons for our experiments with the settings of absence scenarios.
> >
> > We hope you can review our response above and make the best out of this discussion period.
> > If you have any further questions, we are happy to engage in further discussion.

---

> > > ### Author Response · Authors · 2024-11-30
> > > **Reply to Reviewer Z2fY**
> > >
> > > Dear Reviewer Z2fY,
> > >
> > > We really appreciate your effort reviewing our paper.
> > >
> > > During this discussion period, we believe we have addressed your concerns and questions thoroughly. With only one day remaining, we kindly ask if you could confirm your evaluation or share any additional concerns regarding our work.
> > >
> > > Thank you once again for your thoughtful feedback.
> > >
> > > Regards,
> > >
> > > Authors.

---

### Author Response · Authors · 2024-11-24
**Message from Authors to Reviewers**

Dear Reviewers,

Thank you for your valuable comments on our article.

We have carefully revised the manuscript and addressed your feedback in detail. We understand your time is limited, but we sincerely hope you can review our responses. We believe we have resolved your concerns to the best of our ability during this discussion period. If you have any further questions, please feel free to contact us.

We would greatly appreciate it if you could reconsider your evaluation and kindly improve your score if you find our revisions satisfactory.

Thank you again for your time and consideration.

---

### Author Response · Authors · 2024-12-04
**Summary of authors' responses during rebuttal phase and message from the authors**

We thank you for the helpful comments and support with our submission. With this summary, we want to summarize the major contributions, as well as highlight what we have done to fully address all the Reviewers' concerns and questions during this Discussion period.

Regarding our contributions:

- Using a latent reconstruction technique as a means, we can apply pretrained multimodal frameworks to small-scale, multimodal datasets with missing modalities.
- Our work is the first in quantifying and analyzing the uncertainty of a pre-trained multimodal model's output, and its connection to the missing modalities (via the reconstruction process), besides the model' inherent nature themselves. The focus on uncertainty estimation and the use of Error Propagation are unique as acknowledged by Reviewer Z2fY and DTc9.
- We introduce a novel PCC loss to guide the uncertainty estimation process. This loss requires no bias regarding the underlying distribution of predictions but still enforces the desirable characteristic of uncertainty.

Regarding the work we have delivered during this Discussion period:
- We explained the reason behind our choice of baselines for the experiments. We provided additional experiments with more baselines and metrics (suggested by Reviewer 5FYA and dpnx). The result shows that our method produces strong performance and good uncertainty estimation, consistently outperforming all baselines over diverse experiments on multimodal datasets from different domains as acknowledged by Reviewer Z2fY and DTc9.

- We explained that the primary use of uncertainty estimation is a tool for assessing the reliability of results, while the reconstruction technique is a means for leveraging pretrained frameworks efficiently, in response to Reviewer DTc9. We added a method to select a suitable uncertainty threshold in the response to Reviewer 5FYA. We revised the Introduction and Method sections with these clarifications in the revised version.
- We explained that our method is a distribution-free technique that introduces no bias regarding the underlying distribution as shown in the response to Reviewer 5FYA and DTc9. We included this discussion in the Method section.

We believe that our responses have now sufficiently addressed all the concerns of the Reviewers. Our method, SURE, is a highly-performant technique that can be applied to various practical applications with missing data scenarios, especially in accuracy-critical domains such as healthcare or autonomous vehicles.


As we did not hear from all the Reviewers for most of the Rebuttal Phase, we sincerely hope that the Reviewers can take into account the significance and practicality of our work, and especially our responses and clarifications to the Reviewers’ concerns, and consider revising the ratings of our paper accordingly.

Regards,

The Authors.

---

### Note · Authors · 2025-03-25

I have read and agree with the venue's withdrawal policy on behalf of myself and my co-authors.

---

### Meta-Review · Area_Chair_1YoC · 2024-12-18

**Metareview:**

This paper proposes a novel approach for managing missing modalities by incorporating latent space reconstruction into pre-trained multimodal models, as well as uncertainty estimation for both reconstructed modalities and downstream tasks. This is performed adopting a Pearson Correlation-based loss function and applying statistical error propagation in deep model training.

As strong points, this work is recognized to present a sound and original methodology, adaptive to different architectures, provide theoretically insights, and having a good experimental analysis.

However, main concerns relate to missing works when discussing the state of the art, and corresponding lack of comparative analysis, leading to lack of generalization of the experimental results, the tests seem limited to scenarios in which only a selected number of modes is absent, still impeding to clearly assess the generalization of the method, and weak clarity, lack of details or justification of some parts of the methodology. Other issues regarding the experimental analysis and questions about the novelty are also reported.

This paper received all ratings below threshold (5, 5, 5, 5), which are maintained also after rebuttal. This paper was also discussed between the AC and the reviewers, who stuck in their original evaluation, not being fully satisfied of the authors' feedback.
The AC recognizes the good job made by the authors in the rebuttal, but also noted that this was not sufficient to eliminate reviewers' doubts who, unanimously, did not increase their original scores.

For these reasons, this paper is not considered acceptable for publication to ICLR 2025.

**Additional Comments On Reviewer Discussion:**

See above

---

### Decision · Program_Chairs · 2025-01-22

Reject